# Collapse or Thrive? Perils and Promises of Synthetic Data in a Self-Generating World

## Abstract

The increasing presence of AI-generated content on the internet raises a critical question: What happens when generative machine learning models are pretrained on web-scale datasets containing data created by earlier models? Some authors prophesy *model collapse* under a '*replace*' scenario: a sequence of models, the first trained with real data and each later one trained *only on* synthetic data from its preceding model. In this scenario, models successively degrade. Others see collapse as avoidable; in an '*accumulate*' scenario, a sequence of models is trained, but each training uses all real and synthetic data generated so far. In this work, we deepen and extend the study of these contrasting scenarios. First, collapse versus avoidance of collapse is studied by comparing the replace and accumulate scenarios on each of three prominent generative modeling settings; we find the same contrast emerges in all three settings. Second, we study a compromise scenario; the available data remains the same as in the *accumulate* scenario – but unlike *accumulate* and like *replace*, each model is trained using a fixed compute budget; we demonstrate that model test loss on real data is larger than in the *accumulate* scenario, but apparently plateaus, unlike the divergence seen with *replace*. Third, we study the relative importance of cardinality and proportion of real data for avoiding model collapse. Surprisingly, we find a non-trivial interaction between real and synthetic data, where the value of synthetic data for reducing test loss depends on the absolute quantity of real data. Our insights are particularly important when forecasting whether future frontier generative models will collapse or thrive, and our results open avenues for empirically and mathematically studying the context-dependent value of synthetic data.

## 1 Introduction: Model Collapse & Why It Matters

With each passing day, the internet contains increasingly more AI-generated content (Altman, 2024). What is the impact of this for future of deep generative models pretrained on web-scale datasets containing data generated by their predecessors? Previous work forewarned that such model-data feedback loops can exhibit *model collapse*, a phenomenon whereby model performance degrades with each model-fitting iteration such that newer models trend towards useless (Shumailov et al., 2023). This prophecy is deeply concerning because society is increasingly relying on these deep generative models (Bommasani et al., 2022; Reuel et al., 2024; Perrault & Clark, 2024; Kapoor et al., 2024), and model collapse threatens that future models will be made useless as society's current data practices pollute the pretraining data supply.

However, the model collapse literature is replete with different experimental methodologies and different mathematical assumptions of different generative models, with different papers reaching different conclusions (Taori & Hashimoto, 2023; Hataya et al., 2023; Martínez et al., 2023; Shumailov et al., 2023; Alemohammad et al., 2024; Martínez et al., 2023; Bohacek & Farid, 2023; Guo et al., 2024; Bertrand et al., 2024; Briesch et al., 2023; Dohmatob et al., 2024a;b; Gerstgrasser et al., 2024; Seddik et al., 2024; Marchi et al., 2024; Padmakumar & He, 2024; Chen et al., 2024; Ferbach et al., 2024a; Veprikov et al., 2024). These mixed methods and findings make assessing the probability and harm of model collapse difficult.

In this work, we extend the *accumulate* workflow studied in Gerstgrasser et al. (2024) to cover several settings previously not studied in the literature to exhibit its effectiveness over the *replace* workflow.

We begin by testing the following hypothesis – that model collapse emerges in a scenario where models are trained on evolving datasets built by deleting past data en masse; and model collapse is avoided in a scenario where the training datasets instead accumulate all real and synthetic data. We consider whether these claims hold in three new generative model task settings pointed to by recent prominent work; we find the claims hold. We then compare three clear dataset evolution scenarios, focusing on a new middle ground where data accumulate over time but each model is trained under a fixed compute budget; in this middle ground, we find that losses on real test data climb faster than without a compute budget but plateau to lower values than if data are deleted en masse after each model-fitting iteration. These results are consistent across five different generative modeling settings. Lastly, we investigate, in a specific situation, whether the proportion or cardinality of initial real data matters more for preventing model collapse and discover a non-trivial interaction between real and synthetic data: when real data are scarce, an appropriate amount of synthetic data reduces the test loss on real data, whereas when real data are ample, synthetic data increases the test loss on real data. Altogether, our work provides valuable comprehensive insights for predicting likely futures of deep generative models pretrained on web-scale data.

## 2 TESTING TWO MODEL COLLAPSE CLAIMS IN THREE NEW GENERATIVE MODELING SETTINGS

Gerstgrasser et al. (2024) recently made two claims about model collapse:

1. Many previous papers induced model collapse by deleting past data en masse and training largely (or solely) on synthetic data from the latest generative model, and

2. If new synthetic data are instead added to real data, i.e., data accumulate over time, then model collapse is avoided.

These two claims are important for forecasting the future of generative models because, if correct, model collapse is then less likely to pose a realistic threat since accumulating data over time is a more realistic modeling assumption; as a partner at Andreessen Horowitz elegantly explained, deleting data en masse is "not what is happening on the internet. We won't replace the Mona Lisa or Lord of the Rings with AI generated data, but the classics will continue to be part of the training data set" (Appenzeller, 2024).

However, these claims have not been tested in three new generative modeling settings recently introduced by prominent work (Shumailov et al., 2024) for studying model collapse:

1. **Multivariate Gaussian Modeling:** Multivariate Gaussians are repeatedly fit to data and then used to sample new synthetic data for future Gaussian fitting.

2. **Kernel Density Estimation:** Kernel density estimators are repeatedly fit to data and then used to sample new synthetic data for future kernel density estimators.

3. **Supervised Finetuning of Language Models:** Language models are finetuned in a supervised manner and then used to sample new synthetic text for future finetuning.

In this section, we ask and answer:

*In these three new generative modeling settings, is model collapse caused by deleting data en masse and avoided by instead accumulating data?*

In all three settings, we empirically find and (when possible) mathematically prove the answer is yes.

### 2.1 MODEL COLLAPSE IN MULTIVARIATE GAUSSIAN MODELING

We consider repeatedly fitting multivariate Gaussians to data and sampling from the fit Gaussians. We begin with $n$ *real* data drawn from a multivariate Gaussian with mean $\mu^{(0)}$ and covariance $\Sigma^{(0)}$:

$$X_1^{(0)}, ..., X_n^{(0)} \sim_{i.i.d.} \mathcal{N}(\mu^{(0)}, \Sigma^{(0)}).$$

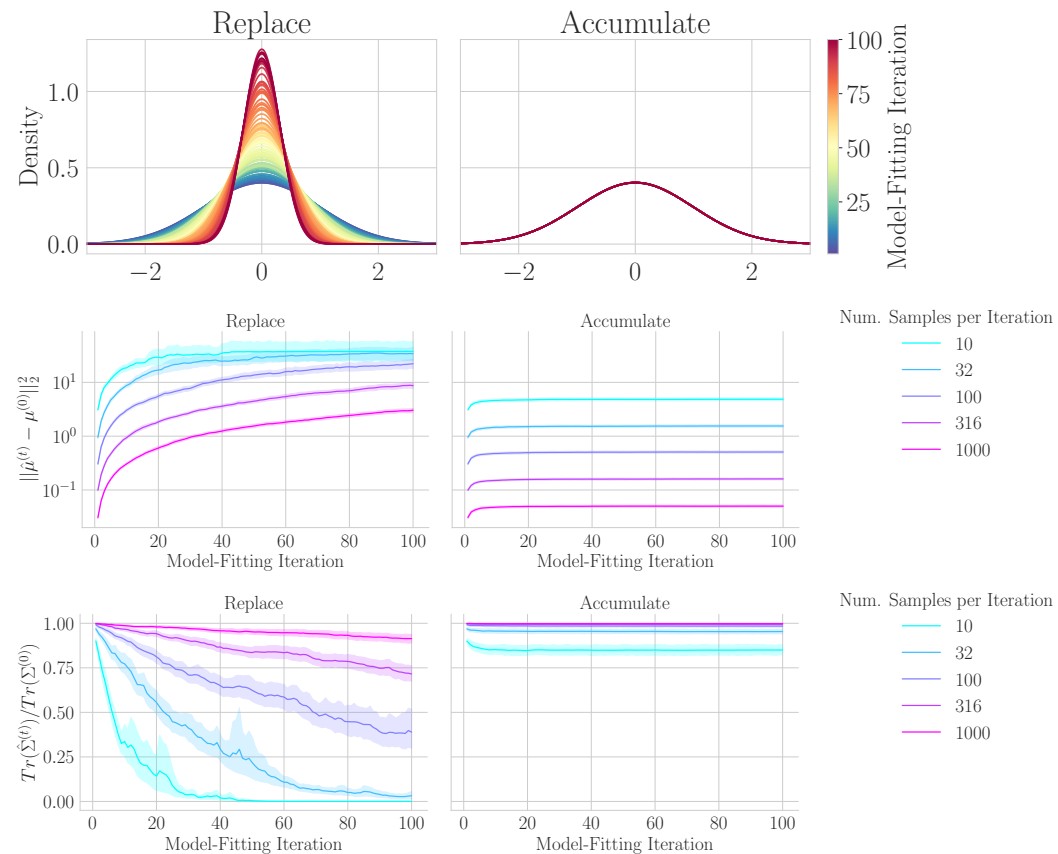

Figure 1: **Model Collapse in Multivariate Gaussian Modeling. Top:** Previous work (Shumailov et al., 2024) proves model collapse occurs if one iteratively fits means and covariances to data and then samples new data from a Gaussian with the fitted parameters (left). We demonstrate that if one doesn't delete all data after each model-fitting iteration - i.e., if data accumulate - then model collapse does not occur (right). Number of Samples Per Iteration: 316. Note: We visualize the fit Gaussians as zero-mean for easy comparison of the fit covariances across model-fitting iterations. **Middle:** If data are replaced, then the empirically fit means drift away from the original data's mean with increasing model-fitting iterations, but if data instead accumulate, then the empirically fit means stabilize. **Bottom:** If data are replaced, then the empirically fit covariances collapse compared to the original data's covariance, but if past data are not discarded, then the fit covariances stabilize quickly and collapse is avoided. Note: Rows 2 and 3 correspond to $d = 31$ dimensional data.

For model fitting, we compute the unbiased mean and covariance of the most recent data:

$$\hat{\mu}_{\text{Replace}}^{(t+1)} \stackrel{\text{def}}{=} \frac{1}{n} \sum_{j=1}^{n} X_j^{(t)} \tag{1}$$

$$\hat{\Sigma}_{\text{Replace}}^{(t+1)} \stackrel{\text{def}}{=} \frac{1}{n-1} \sum_{j=1}^{n} (X_j^{(t)} - \hat{\mu}_{\text{Replace}}^{(t+1)})(X_j^{(t)} - \hat{\mu}_{\text{Replace}}^{(t+1)})^T \tag{2}$$

For model sampling, we sample $m$ new synthetic data using the fit Gaussian parameters:

$$X_1^{(t)}, ..., X_n^{(t)} \,\Big|\, \hat{\mu}_{\text{Replace}}^{(t)}, \hat{\Sigma}_{\text{Replace}}^{(t)} \quad \sim_{i.i.d.} \quad \mathcal{N}(\hat{\mu}_{\text{Replace}}^{(t)}, \hat{\Sigma}_{\text{Replace}}^{(t)}). \tag{3}$$

Under the above data-model feedback loop, Shumailov et al. (2024) prove that

$$\hat{\Sigma}_{\text{Replace}}^{(t+1)} \stackrel{a.s.}{\to} 0 \quad ; \quad \mathbb{E}[\mathbb{W}_2^2(\mathcal{N}(\hat{\mu}_{\text{Replace}}^{(t+1)}, \hat{\Sigma}_{\text{Replace}}^{(t+1)}), \mathcal{N}(\mu^{(0)}, \Sigma^{(0)}))] \to \infty \text{ as } t \to \infty, \tag{4}$$

where $\mathbb{W}_2$ denotes the Wasserstein-2 distance. This result states that the fit covariance will collapse to 0 and that the Wasserstein-2 distance will diverge as this model-data feedback loop unfolds. Note

that the Wasserstein-2 distance diverges not because the covariance collapses to $0$ but because the distance between the $t$-th fit mean $\hat{\mu}_{\text{Replace}}^{(t)}$ and the true mean $\mu^{(0)}$ diverges.

However, *this result assumes that all data are deleted after each model-fitting iteration.* As discussed above, this assumption is likely unrealistic because society does not delete earlier content from the internet and replace it with new model-generated content after fitting each state-of-the-art model. What happens if data instead *accumulate* across model-fitting iterations? To study this, we instead consider fitting to all previous real and synthetic data:

$$\hat{\mu}_{\text{Accumulate}}^{(t+1)} \overset{\text{def}}{=} \frac{1}{n(t+1)} \sum_{i=0}^{t} \sum_{j=1}^{n} X_j^{(i)} \tag{5}$$

$$\hat{\Sigma}_{\text{Accumulate}}^{(t+1)} \overset{\text{def}}{=} \frac{1}{n(t+1)-1} \sum_{i=0}^{t} \sum_{j=1}^{n} (X_j^{(i)} - \hat{\mu}_{\text{Accumulate}}^{(t+1)})(X_j^{(i)} - \hat{\mu}_{\text{Accumulate}}^{(t+1)})^T \tag{6}$$

Data are then sampled using these fit Accumulate parameters rather than the fit Replace parameters.

Empirically, we find that deleting all data after each model-fitting iteration causes model collapse (Fig. 1 Left), whereas accumulating data across model-fitting iterations prevents model collapse (Fig. 1 Right). More specifically, we find that if data are deleted, the squared error between the fit mean $\hat{\mu}_{\text{Replace}}^{(n)}$ and the initial mean $\mu^{(0)}$ diverges (Fig. 1, Middle Left), and the fit covariance $\hat{\Sigma}_{\text{Replace}}^{(n)}$ relative to the initial covariance $\Sigma^{(0)}$ collapses to $0$ (Fig. 1, Bottom Left), as measured by the ratio between the trace of $\hat{\Sigma}^{(t)}$ and the trace of $\Sigma^{(0)}$. In contrast, if data accumulate, the squared error between the fit mean and the initial mean plateaus quickly (Fig. 1, Middle Right), as does the fit covariance relative to the initial covariance (Fig. 1, Bottom Right).

Additionally, in the univariate case, we mathematically characterize the limit distribution:

**Theorem 1.** *For notational efficiency, for a univariate Gaussian, let $\hat{\mu}^{(t)}$ and $\hat{\sigma}^{(t)}$ denote $\hat{\mu}_{Accumulate}^{(t)}$ and $\hat{\Sigma}_{Accumulate}^{(t)}$. Suppose that the mean and covariance are updated as in Eqns. 5 and 6. Then*

$$\mathbb{E}\left(\sigma_t^2\right) = \sigma_0^2 \cdot \prod_{k=1}^{t} \left(1 - \frac{1}{k^2 n}\right) \quad \xrightarrow{t \to \infty} \quad \sigma_0^2 \cdot \left(\frac{\sin(\pi/\sqrt{n})}{\pi/\sqrt{n}}\right) \tag{7}$$

$$\mathbb{E}[(\mu_t - \mu_0)^2] = \sigma_0^2 \cdot \left(1 - \prod_{k=1}^{t} \left(1 - \frac{1}{k^2 n}\right)\right) \quad \xrightarrow{t \to \infty} \quad \sigma_0^2 \cdot \left(1 - \frac{\sin(\pi/\sqrt{n})}{\pi/\sqrt{n}}\right). \tag{8}$$

See Appendix Sec. A for the proof. This reveals two key differences when data accumulate: the covariance no longer collapses, and the mean no longer diverges, meaning model collapse is mitigated.

## 2.2 Model Collapse in Kernel Density Estimation

We next turn to the second generative modeling setting for studying model collapsed introduced by Shumailov et al. (2024): kernel density estimation (KDE). Similar to multivariate Gaussian modeling, we begin with $n$ real data points drawn from an initial probability distribution $p^{(0)}$: $X_1^{(0)}, ..., X_n^{(0)} \sim_{i.i.d.} p^{(0)}$. We then iteratively fit KDEs to the data and sample new synthetic data from these estimators. In the Replace setting, we fit the KDE to $n$ data samples from the most recently fit model, whereas in the Accumulate setting, we fit the KDE to all data points from all previous iterations, with the number of points growing linearly as $n(t+1)$:

$$\hat{p}_{\text{Replace}}^{(t+1)}(x) \overset{\text{def}}{=} \frac{1}{nh} \sum_{j=1}^{n} K\left(\frac{x - X_j^{(t)}}{h}\right) \tag{9}$$

$$\hat{p}_{\text{Accumulate}}^{(t+1)}(x) \overset{\text{def}}{=} \frac{1}{nh(t+1)} \sum_{i=0}^{t} \sum_{j=1}^{n} K\left(\frac{x - X_j^{(i)}}{h}\right) \tag{10}$$

where $K$ is the kernel function and $h$ is the bandwidth parameter. We consider a standard Gaussian kernel. For sampling, at each iteration, we draw $n$ new synthetic data points from the fitted kernel

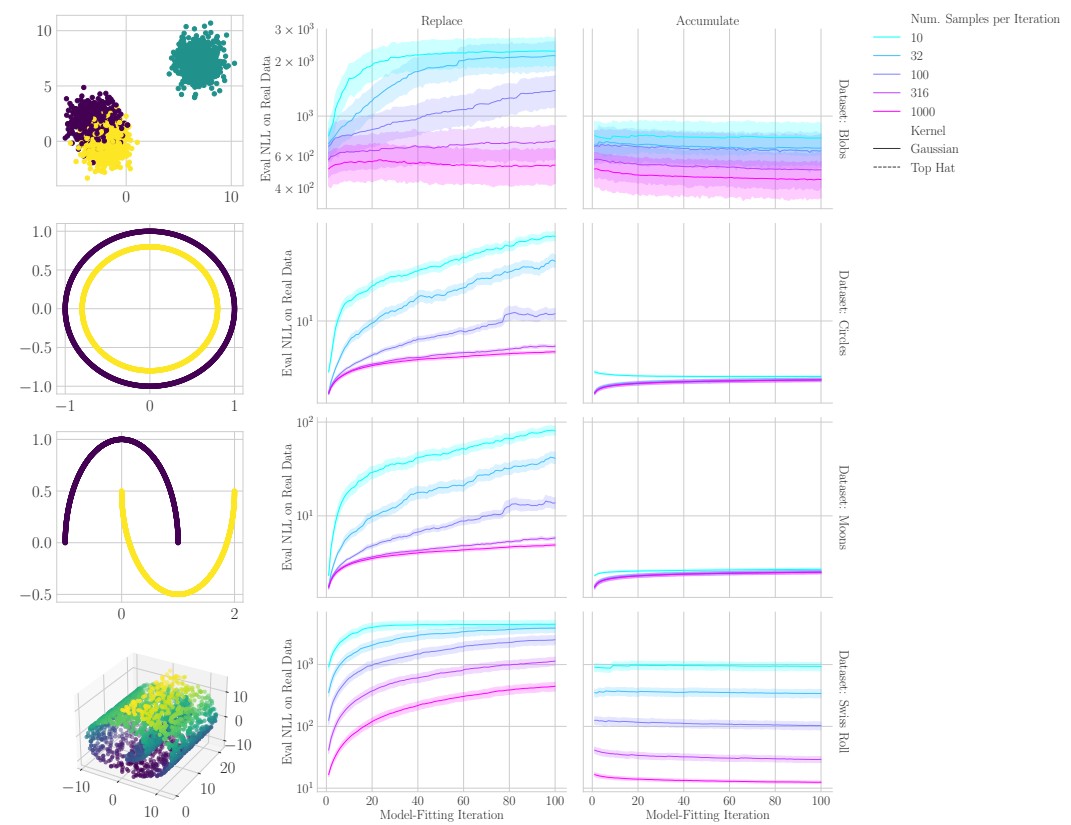

Figure 2: **Model Collapse in Kernel Density Estimation.** Left: We consider 4 standard datasets from `sklearn`: Blobs, Circles, Moons and Swiss Roll. Center: For all four datasets, deleting data en masse causes the negative log likelihoods (NLL) of real test data to increase with each model-fitting iteration. Right: For all four datasets, accumulating data avoids model collapse. Interestingly, for specific pairs of datasets and number of samples per iteration, training on real and accumulating synthetic data can yield *lower loss on real test data* than training on real data alone.

density estimators. We evaluate the performance using the negative log-likelihood (NLL) on real held-out test data; lower NLL indicates better performance. For data, we use four standard synthetic datasets from `sklearn` (Pedregosa et al., 2011): blobs, circles, moons, and swiss roll.

We again observe the same general difference between replacing data and accumulating data (Fig. 2): replacing data causes a rapid increase in NLL as the number of model-fitting iterations increases, indicating that the KDEs are becoming increasingly poor at modeling the true underlying distribution. In contrast, when data accumulate across model-fitting iterations, we observe that the NLL remains relatively stable, suggesting that accumulating data helps maintain the quality of the KDEs.

Despite the apparent empirical similarities to the iterative Gaussian fitting shown in Figure 1, Gaussian KDEs are theoretically distinct. Unlike Gaussian fitting, regardless of whether we accumulate or replace, the test NLL for a Gaussian KDE asymptotically diverges in theory. There are two interesting caveats: (1) when one begins with a small bandwidth, iteratively fitting KDEs can cause the NLL to initially decrease before it diverges due to the effective kernel bandwidth increasing with model-fitting iterations; (2) although accumulating data causes the NLL to diverge asymptotically, this occurs at a rate so glacial that it doesn't pose a practical concern. If one wishes to prevent the eventual divergence, one can do so by fitting at each iteration with the optimal bandwidth for the number of data, which should be of the form $c(in)^{-1/5}$ in the $i$th model-fitting iteration as long as data accumulates at a constant rate. Practically speaking, one chooses the bandwidth for KDEs based on the number and characteristics of the data, implying that conscientious practitioners should never witness severe model collapse for KDEs in the accumulate case. For details, see Appendix Sec. B.

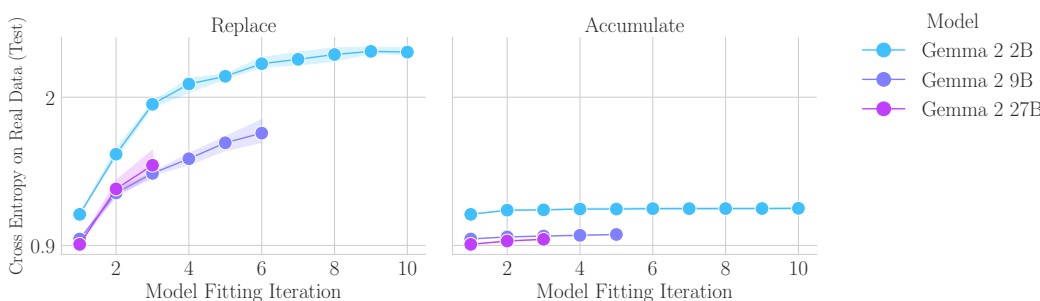

Figure 3: **Model Collapse in Supervised Finetuning of Language Models.** Finetuning Google's Gemma2 2B on Nvidia's HelpSteer 2 dataset demonstrates that model collapse occurs if previous data are replaced after each model-fitting iteration (left), whereas model collapse is avoided if new synthetic data instead accumulate with previous real and synthetic data (right).

We also note a surprising discovery: accumulating data can yield NLLs that decrease with additional model-fitting iterations, meaning that training on real and synthetic data yields lower test loss on real data than training on real data alone. While synthetic data has been shown valuable elsewhere, e.g., Jain et al. (2024), we were surprised to discover this behavior in such a simple setting. This behavior is analogous to Mobahi et al. (2020), which demonstrated how self-distillation of linear models can initially improve model performance by acting as increasing regularization in Hilbert space, but if too many iterations take place, the predictor is regularized towards 0 and performance deteriorates.

### 2.3 MODEL COLLAPSE IN SUPERVISED FINETUNING OF LANGUAGE MODELS

We now turn to the third setting for studying model collapse introduced by Shumailov et al. (2024): supervised finetuning of language models. We begin with an instruction following dataset – Nvidia's HelpSteer2 (Wang et al., 2024) – and finetune a language model before sampling new text data from it. We choose Google's Gemma2 2B model (Team et al., 2024) because it is high performing and relatively small. For Replace, we fine-tune the $n$-th language model only on data generated by the $(n - 1)$ language model. For Accumulate, we instead fine-tune the $n$-th language model on the starting real data plus all the synthetic data sampled from all previous models; thus, the amount of data for Replace is constant $\sim 12.5k$, whereas the amount of data for Accumulate grows linearly $\sim 12.5k * t$. Consistent with our results and with Gerstgrasser et al. (2024), we find that deleting data after each iteration leads to collapse whereas accumulating data avoids collapse (Fig. 3).

## 3 MODEL COLLAPSE UNDER A FIXED COMPUTE BUDGET

Thus far, we have focused on two data paradigms: Replace and Accumulate. As discussed in Sec. 2, Replace is unlikely to be an faithful model of reality because we do not delete the internet after pretraining each model. But one might argue that Accumulate is similarly unfaithful because Accumulate requires that every new model is trained on (linearly) more data and thus requires more compute than its predecessor. Whether this criticism is valid in practice is unclear, since newer models *are* trained on increasing data (e.g., 1.4T tokens for Llama 1, 2T tokens for Llama 2, 15T tokens from Llama 3) and increasing GPUs (e.g., 2k GPUs for Llama1, 4K for Llama2, 16k for Llama3 (Goyal, 2024)). Nevertheless, for the sake of understanding the space of possible outcomes and predicting likely outcomes for future generative models, we ask and answer:

> *Does model collapse occur when data accumulate but models are trained under a*
> *fixed compute budget?*

We call this data paradigm *Accumulate-Subsample* because data accumulate but are then subsampled to ensure constant data and thus constant compute at each model-fitting iteration. To study whether model collapse occurs in Accumulate-Subsample, we use the same three generative modeling settings we've studied (multivariate Gaussian modeling, supervised finetuning of language models and kernel density estimation) plus two new generative modeling settings studied by prior work (Mobahi et al.,

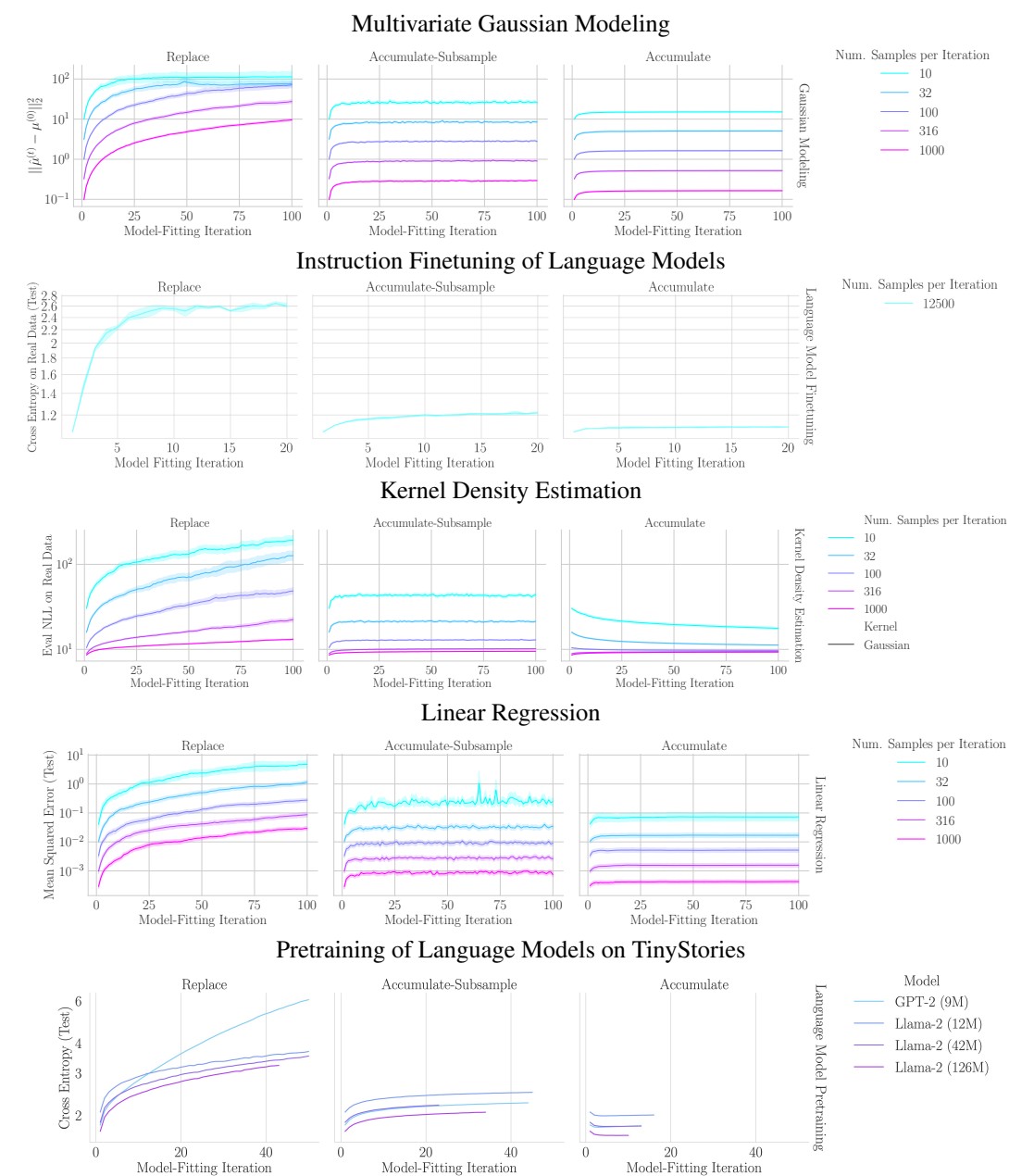

Figure 4: **Model Collapse Under a Fixed Compute Budget.** We compare deleting data after each model-fitting iteration (Replace) and accumulating data after each iteration (Accumulate) with a new fixed-compute data paradigm Accumulate-Subsample. In Accumulate-Subsample, real and synthetic data accumulate but are then subsampled so that each model is trained on a constant number of data. Accumulate-Subsample's test loss on real data deteriorates more quickly than Accumulate's loss but more slowly than Replace's loss, and frequently converges, albeit to a higher plateau than Accumulate. These results hold across five settings: multivariate Gaussian modeling, language model instruction finetuning, kernel density estimation, linear regression and language model pretraining.

2020; Dohmatob et al., 2024a; Gerstgrasser et al., 2024): linear regression and pretraining language models on a GPT3.5/GPT4-generated dataset of kindergarten-level text (Eldan & Li, 2023).

To explain how linear regression can be used as a generative model, we briefly here and direct the reader to prior work (Mobahi et al., 2020; Dohmatob et al., 2024a; Gerstgrasser et al., 2024) for a

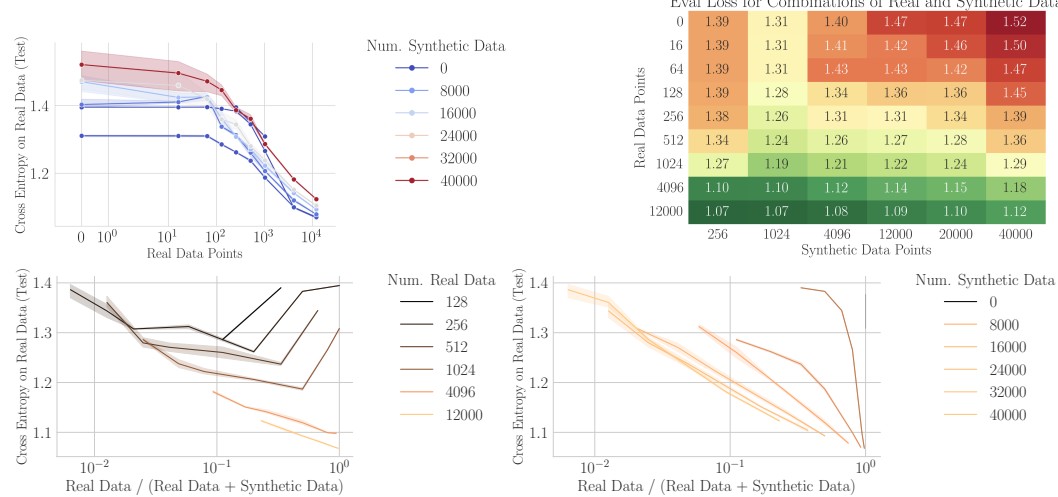

Figure 5: **The Value of Synthetic Data in Supervised Finetuning of Language Models.**
Finetuning Google's Gemma 2 2B on Nvidia's HelpSteer 2 dataset on different combinations of real and synthetic data demonstrates that loss grows with the number of synthetic data. Our results suggest that the test loss depends on both the proportion (p-value$= 3.84 \times 10^{-16}$) and the cardinality (p-value$= 3.54 \times 10^{-8}$) of real data. We plot the test loss against the number of real datapoints in the training set (top left). The hue represents the number of synthetic datapoints. Additionally, we display a heatmap demonstrating the effect of the number of real and synthetic datapoints on test loss (top right). We provide a graph of the test loss versus the fraction of real data, where the hue represents the cardinality of the real data (bottom left). Finally, we plot the test loss against the fraction of real datapoints, where the hue represents the number of synthetic datapoints (bottom right).

more thorough description. We begin with our real covariates $X \in \mathbb{R}^{n \times d}$ and true linear relationship $w^{(0)}$. Initializing $\hat{w}^{(0)} = w^{(0)}$, we sample the regression targets as:

$$y^{(t)} \stackrel{\text{def}}{=} X\hat{w}^{(t)} + E^{(t)} \quad ; \quad E^{(t)} \sim \mathcal{N}(0, \sigma^2 I_d) \tag{11}$$

Assuming $X^T X$ is full rank, e.g., $n \gg d$, we fit the next linear model using ordinary least squares:

$$\hat{w}^{(t+1)} \stackrel{\text{def}}{=} (X^T X)^{-1} X^T y^{(t)} \tag{12}$$

Following Gerstgrasser et al. (2024), we additionally pretrain sequences of small variants of common large language models – GPT (Radford et al., 2019; Brown et al., 2020) and Llama (Touvron et al., 2023a;b) – on TinyStories (Eldan & Li, 2023), a synthetic dataset of simple short stories; this combination of models, parameters and data was chosen to faithfully study model collapse in as realistic a setting as possible, subject to our limited computational budget.

Across all five generative modeling settings, we find that Accumulate-Subsample's test loss on real data lies between the test losses of Replace and Accumulate (Fig. 4 Center). Specifically, Accumulate-Subsample (Fig. 4 center) exhibits higher test loss than Accumulate (Fig. 4, Right) but lower test loss than Replace (Fig. 4 Left), showing that the fixed compute budget imposes some cost. In a qualitative difference, test losses on real data typically plateaus for both Accumulate-Subsample and Accumulate, whereas test losses for Replace typically diverge in an apparently unbounded manner. These results collectively tell a consistent story: under more realistic conditions, where data accumulate and compute is bounded, model performance on real test data is unlikely to diverge.

## 4 CARDINALITY OF REAL DATA VS PROPORTION OF REAL DATA IN MITIGATING MODEL COLLAPSE

We conclude by turning to a question asked by Gerstgrasser et al. (2024) that, to the best of our knowledge, remains open:

*Which matters more for avoiding model collapse: the cardinality of real data or the proportion of real data? Relatedly, how does the value of synthetic data for reducing test loss on real data depend on the amount of real data?*

These questions are highly pertinent to researchers sampling from web-scale data in order to pretrain or finetune language models. We conduct our investigation of this question as follows: First, we perform SFT on the HelpSteer2 dataset for Google's Gemma 2 2B model. We sample 100k completions from the finetuned model and filter for those that are fewer than 512 tokens in length. This leaves us with over 55,000 remaining completions. We aggregate datasets containing various numbers of real and synthetic synthetic data, which are given in Figure 5, and perform SFT on these datasets starting from the original Gemma 2B model. We record and display the final test loss from this process.

This experiment provides several insights. First, both the number and proportion of real data have an impact on the test loss following SFT. To assess this, we first transformed the number of real datapoints $n$ as $\frac{1}{n^{1/2}}$, in keeping with intuitions from classical statistics on how the log likelihood scales with the number of data points. Then, based on observation of the data, we computed

$$\log\left(\frac{\text{real data}}{\text{real data} + \text{synthetic data}}\right)$$

to best capture the relationship between the fraction of real data and the log likelihood. We measured $R^2$ values of $0.59$ for the transformed number of real data and $0.34$ for the proportion of real data. We then computed $F$-statistics for the one-term versus two term models involving each of these covariates, which gave us $p$-values of $6.9 \times 10^{-25}$ and $4.6 \times 10^{-25}$. These statistics suggest that both the proportion and the cardinality of real data have a statistically significant effect on the test loss, and explain a sizable fraction of the variance in the test loss.

Second, we find a difference in the effect that synthetic data has on test loss in high versus low real data regimes. In our experiments, when the number of real data is 1024 or lower, we find that there is an *small but non-zero amount of synthetic data that improves the test loss when it is included*. This suggests that practitioners fine-tuning with insufficient amounts of real data should consider supplementing with synthetic data to improve model quality. On the other hand, when real data are plentiful, we find that more synthetic data almost always harms final model quality when the number of real data is held constant. In some cases, datasets containing only real data prove to be more valuable than datasets that contain ten times more real data mixed with synthetic data.

Although these results are preliminary, they raise interesting questions about the role of synthetic data in SFT that merit exploration. In some of our experiments, we achieve better results by removing all synthetic data from the training set than by doubling the amount of real data. When constructing datasets subject to cost constraints, these results suggest that removing synthetic or low-quality data can sometimes bring more value than collecting greater volumes high-quality data.

## 5 RELATED WORK

The limitations of using AI-generated images to train other image models have been well-documented since 2022 (Hataya et al. (2023)). Shumailov et al. (2023) initially sounded alarms about synthetic data for training language models by showing that a model trained repeatedly on its own outputs exhibits severely denigrated quality. This theory and empirical work was quickly extended to many new settings (Alemohammad et al. (2024); Bertrand et al. (2024); Dohmatob et al. (2024b;a); Marchi et al. (2024)). The phenomenon that Shumailov identified as "model collapse" still does not have a universally agreed upon, rigorous definition. Shumailov classified model collapse as a "degenerative process affecting generations of learned generative models, in which the data they generate end up polluting the training set of the next generation." Dohmatob et al. (2024b) examine model collapse as an alteration of scaling law curves when training on synthetic as opposed to real data. In their theory sections, Shumailov et al. (2024) and Gerstgrasser et al. (2024) explore model collapse by asking when certain models exhibit divergent test loss after multiple iterations of training. In this paper, we take model collapse by its literal meaning: that model performance deteriorates catastrophically when models are trained on synthetic data.

Within the model collapse literature, a variety of data dynamics have been studied, which vary in how "real" data is discarded or retained, how "synthetic" data is generated, and how each is (or is

not) incorporated into future training sets (Martínez et al. (2023); Mobahi et al. (2020); Dohmatob et al. (2024a)). A common feature of many of these is that at least some real data is discarded, often because total dataset size is kept constant across model-fitting iterations. However, Gerstgrasser et al. (2024) note that this may not be representative of real-world dynamics, and that model collapse is avoided when data accumulates. What is not clear, however, is whether this claim holds universally, including in the specific settings studied in other prior work. We help close this gap by extending Gerstgrasser's empirical and theoretical analysis to several of these settings.

Where model collapse can be seen as studying a worst-case scenario, it has also been observed that *some* kinds of synthetic data have a positive effect. Dohmatob et al. (2024b) and Jain et al. (2024) find that certain amounts of synthetic data can improve model performance, and Ferbach et al. (2024b) suggest that with curation, self-consuming loops can improve alignment with human preferences. A growing literature on how to filter and harness synthetic data has achieved impressive results on a variety of benchmarks (Zelikman et al. (2024); Li et al. (2024a); Yang et al. (2024)), raising interesting questions about the limits of when unfiltered synthetic data can help. In this vein, we answer a question posed by Gerstgrasser et al. (2024): does the proportion or the raw amount of real data in a mixed training set have a greater impact on test loss? In the process, we find that small amounts of synthetic data can improve test loss when real data is scarce.

## 6 DISCUSSION

Our work sought extend understanding of model collapse in the replace and accumulate workflows. We demonstrated in three new generative modeling settings that accumulating data over time avoids model collapse, whereas replacing data over time induces model collapse. We then demonstrated in five generative modeling settings that even when each model is trained on a fixed compute budget with a mixture of real and synthetic data, model performance does deteriorate more, but still tends to plateau. The consistency of these results across different model types and datasets suggests that *this distinction is a general phenomenon, and is not specific to any particular model, dataset, or learning algorithm.* Lastly, we explored the value of synthetic data for reducing the test loss on real data and found two different regimes: when real data are plentiful, synthetic data is harmful, but when real data are scarce, there exists an optimal amount of synthetic data that are helpful.

In our view, the data paradigm in which synthetic data accumulates from a host of models in conjunction with a constant influx of real-world data is more realistic. Under such dynamics, where new synthetic data are added to existing real and synthetic data, model collapse appears unlikely. Our experiments take a pessimistic viewpoint, in the sense that our experiments pay no attention to the quality of data, whereas in practice, engineers heavily filter data based on various indicators of data quality, e.g., (Brown et al., 2020; Lee et al., 2023; Wettig et al., 2024; Penedo et al., 2024; Li et al., 2024b; Sachdeva et al., 2024); for a recent review, see Albalak et al. (2024).

## 7 FUTURE DIRECTIONS

An especially interesting future direction is how to combine synthetic data generation with filtering techniques to enable performant and efficient pretraining at scale using synthetic data. As we saw in kernel density estimation (Fig. 2) and in language model pretraining on TinyStories (Fig. 4), training on accumulating real and synthetic data can yield lower loss on real test data than training on real data alone. Identifying under what conditions, and why, this is possible is a tantalizing prospect.

Our results in Section 4 suggest that removing low-quality synthetic data from model training sets *can improve test loss more than gathering additional high-quality data*. Developing efficient identification and removal techniques for detrimental data would streamline the model fine-tuning process and produce better alignment.

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

## A  ITERATIVE GAUSSIAN MODEL FITTING: MATHEMATICAL RESULTS AND PROOFS

### A.1  SETUP

**Lemma 2.** *Using the notation of Theorem 1, we can express $\mu_t = \sum_{r=1}^{t} \sigma_{r-1} \frac{\overline{z_r}}{r} + \mu_0$.*

*Proof.* Note that $X_{i,t} = \mu_{t-1} + \sigma_{t-1} z_{i,t}$, where $z_{i,t} \sim \mathcal{N}(0,1)$. Therefore,

$$
\begin{aligned}
\mu_t &= \frac{1}{nt} \sum_{r=1}^{t} \sum_{i=1}^{n} X_{i,r} \\
&= \frac{t-1}{t} \mu_{t-1} + \frac{\mu_{t-1}}{t} + \sigma_{t-1} \frac{\overline{z_t}}{t} \\
&= \mu_{t-1} + \sigma_{t-1} \frac{\overline{z_t}}{t}.
\end{aligned}
$$

Therefore, $\mu_t = \sum_{r=1}^{t} \sigma_{r-1} \cdot \frac{\overline{z_r}}{r} + \mu_0$. $\qquad\qquad\square$

**Lemma 3.** *Under the setup described in Theorem 1, $\mathbb{E}[\frac{\sigma_t^2}{\sigma_0^2}] = \prod_{k=1}^{t} \left(1 - \frac{1}{nk^2}\right) \xrightarrow{t \to \infty} \frac{\sin(\pi/\sqrt{n})}{\pi/\sqrt{n}}$.*

*Proof.* Using the recursive expression for $\mu_t$ in Lemma 2, we can rewrite

$$
\begin{aligned}
\sigma_t^2 &= \frac{1}{nt} \sum_{r=1}^{t} \sum_{i=1}^{n} \left(X_{i,r} - \mu_t\right)^2 \\
&= \frac{1}{nt} \sum_{r=1}^{t} \sum_{i=1}^{n} \left(X_{i,r} - \overline{X_r} + \overline{X_r} - \mu_t\right)^2 \\
&= \frac{1}{nt} \sum_{r=1}^{t} \left(\sum_{i=1}^{n} \left(X_{i,r} - \overline{X_r}\right)^2 + n(\overline{X_r} - \mu_t)^2\right) \\
&= \frac{1}{t} \sum_{r=1}^{t} \left(\sigma_{r-1}^2 S_r^2 + (\mu_{r-1} + \sigma_{r-1}\overline{z_r} - \mu_t)^2\right).
\end{aligned}
$$

In the last line, we define $S_r^2 = \sum_{i=1}^{n}(z_{i,r} - \overline{z_r})^2$. The term

$$
(\mu_{r-1} + \sigma_{r-1}\overline{z_r} - \mu_t)^2 = \left(\sigma_{r-1}\overline{z_r} - \sum_{k=r}^{t} \sigma_{k-1} \cdot \frac{\overline{z_k}}{k}\right)^2,
$$

so

$$
\begin{aligned}
\sigma_t^2 &= \frac{1}{t} \sum_{r=1}^{t} \left(\sigma_{r-1}^2 S_r^2 + \left(\sigma_{r-1}\overline{z_r} - \sum_{k=r}^{t} \sigma_{k-1} \frac{\overline{z_k}}{k}\right)^2\right) \\
\Rightarrow t\sigma_t^2 &= \sum_{r=1}^{t} \left(\sigma_{r-1}^2 S_r^2 + \left(\sigma_{r-1}\overline{z_r}\left(1 - \frac{1}{r}\right) - \sum_{k=r+1}^{t} \sigma_{k-1} \frac{\overline{z_k}}{k}\right)^2\right).
\end{aligned}
$$

We now compute the conditional expectations of the terms in this sum. Where $\mathcal{F}_i$ denotes the $i$th filtration,

$$
\mathbb{E}[\sigma_{r-1}^2 S_r^2 | \mathcal{F}_{t-1}] = \begin{cases} \sigma_{r-1}^2 S_r^2 & r < t \\ \sigma_{t-1}^2 \cdot \left(\frac{n-1}{n}\right) & r = t. \end{cases}
$$

For $r = t$, we find that

$$
\mathbb{E}\left[\left(\sigma_{r-1}\overline{z_r} \cdot \left(1 - \frac{1}{r}\right) - \sum_{k=r+1}^{t} \sigma_{k-1} \cdot \frac{\overline{z_k}}{k}\right)^2 | \mathcal{F}_{t-1}\right] = \sigma_{t-1}^2 \left(1 - \frac{1}{t}\right) \cdot \frac{1}{n}.
$$

On the other hand, when $r < t$,

$$\mathbb{E}\left[\left(\sigma_{r-1}\overline{z_r} \cdot \left(1 - \frac{1}{r}\right) - \sum_{k=r+1}^{t-1} \sigma_{k-1} \cdot \frac{\overline{z_k}}{k} - \sigma_{t-1} \cdot \frac{\overline{z_t}}{t}\right)^2 \bigg| \mathcal{F}_{t-1}\right]$$

$$= \sigma_{t-1}^2 \cdot \frac{1}{t^2} \cdot \frac{1}{n} + \left(\sigma_{r-1}\overline{z_r} \cdot \left(1 - \frac{1}{r}\right) - \sum_{k=r+1}^{t-1} \sigma_{k-1} \cdot \frac{\overline{z_k}}{k}\right)^2.$$

Therefore,

$$\mathbb{E}[t\sigma_t^2 | \mathcal{F}_{t-1}] = (t-1)\sigma_{t-1}^2 + \sigma_{t-1}^2 \cdot \left(1 - \frac{1}{n}\right) + \sigma_{t-1}^2 \cdot \left(\frac{t-1}{t}\right) \cdot \left(\frac{1}{n}\right) + \sigma_{t-1}^2 \cdot \left(1 - \frac{1}{t}\right)^2 \cdot \left(\frac{1}{n}\right)$$

$$= \sigma_{t-1}^2 \left(t - 1 + 1 - \frac{1}{n} + \frac{1}{tn} - \frac{1}{t^2n} + \frac{1}{n} - \frac{2}{tn} + \frac{1}{t^2n}\right)$$

$$= \sigma_{t-1}^2 \left(t - \frac{1}{tn}\right).$$

It follows that

$$\mathbb{E}[\sigma_t^2 | \mathcal{F}_{t-1}] = \sigma_{t-1}^2 \left(1 - \frac{1}{t^2n}\right) < \sigma_{t-1}^2$$

for all $t$. Thus, $\{\sigma_t^2\}_t$ is a supermartingale, and

$$\sigma_t^2 \xrightarrow{a.s.} \sigma_\infty^2$$

because $\sigma_t^2$ is bounded below by $0$. Therefore, we still have convergence. Next, letting $m_t = \mathbb{E}[\sigma_t^2]$, we have

$$m_t = m_{t-1}\left(1 - \frac{1}{t^2n}\right) = \cdots = \sigma_0^2 \prod_{k=1}^{t} \left(1 - \frac{1}{k^2n}\right),$$

so

$$\mathbb{E}[\sigma_t^2] = \sigma_0^2 \prod_{k=1}^{\infty} \left(1 - \frac{1}{k^2n}\right). \tag{13}$$

By a theorem of Euler, this is equal to

$$\sigma_0^2 \frac{\sin(\pi/\sqrt{n})}{\pi/\sqrt{n}}. \tag{14}$$

$\square$

Observe that by performing a variable replacement and using L'Hospital's rule, it is clear that $\lim_{n\to\infty} \mathbb{E}[\sigma_t^2] = \sigma_0^2$.

Finally, we are able to compute $\mathbb{E}[(\mu_t - \mu_0)^2]$.

**Corollary 4.** *The expected error in the mean*

$$\mathbb{E}[(\mu_t - \mu_0)^2] = \sigma_0^2 \left(1 - \prod_{k=1}^{t} \left(1 - \frac{1}{k^2n}\right)\right). \tag{15}$$

*Proof.* Using the recursion from Lemma 2 and the expression for the variance in Lemma 6, we can rewrite

$$\mathbb{E}[(\mu_t - \mu_0)^2] = \sum_{k=1}^{t} \frac{\mathbb{E}[\sigma_{k-1}^2]}{nk^2}$$

$$= \sigma_0^2 \sum_{k=1}^{t} \frac{1}{k^2 n} \prod_{\ell=1}^{k-1} \left(1 - \frac{1}{\ell^2 n}\right)$$

$$= \sigma_0^2 \sum_{k=1}^{t} \left( \prod_{\ell=1}^{k-1} \left((1 - \frac{1}{\ell^2 n}) - \prod_{\ell=1}^{k} \left(1 - \frac{1}{\ell^2 n}\right)\right)\right)$$

$$= \sigma_0^2 \left(1 - \prod_{k=1}^{t} \left(1 - \frac{1}{k^2 n}\right)\right).$$

$\square$

Therefore,

$$\lim_{t \to \infty} \mathbb{E}[(\mu_t - \mu_0)^2] = \sigma_0^2 \left(1 - \frac{\sin(\pi/\sqrt{n})}{\pi/\sqrt{n}}\right).$$

## B  ITERATIVE KDE FITTING: MATHEMATICAL RESULTS AND PROOFS

In this section, we prove that the NLL diverges when iteratively fitting KDE's regardless of whether one accumulates or replaces data from previous iterations.

**Theorem 5.** *In the replace setting described in Section 2.2, as long as one holds the bandwidth constant, the NLL asymptotically diverges.*

*Proof.* Define $f_0$ as the density function for the data distribution from which the original data $x_1, ..., x_n$ are sampled. Define $K_h$ to be the Gaussian kernel function with fixed bandwidth $h$. One can rewrite the fitted distribution at iteration $t$ as

$$D_t = K_h * D_{t-1}$$

where $*$ denotes the standard convolution of densities.

By a simple recursion, it is clear that $D_t = K^{*t} * D_0$. When two Gaussian kernels with bandwidths $a$ and $b$ are convolved, a basic calculation shows that the resulting effective bandwidth is $\sqrt{a^2 + b^2}$. Consequently, by an inductive argument, the effective bandwidth of $K^{*t}$ is $h\sqrt{t}$. Therefore,

$$\lim_{t \to \infty} K^{*t} * D_0 = \lim_{t \to \infty} K_{h\sqrt{t}} * D_0 = 0$$

because as the bandwidth goes to $\infty$, the likelihood of any point goes to $0$. Hence, regardless of the choice of test data, the negative log likelihood diverges to $-\infty$. $\square$

The same conclusion holds when one accumulates rather than subsampling data:

**Theorem 6.** *For any non-trivial kernel (i.e. a kernel whose Fourier transform is not 1), 2.2, the NLL diverges.*

*Proof.* We adopt the same notation as in Theorem 5, except this time $K$ denotes a general kernel $K$ that doesn't necessarily need to be Gaussian. In this instance, it is more convenient to work in frequency space, where convolution in probability space corresponds to multiplication.

Define $\varphi_0$ as the Fourier transform (FT) of $f_0$, also called the characteristic function. Let $\kappa$ denote the FT of $K$. Then

$$\varphi_t = \kappa \cdot \varphi_{t-1}$$

where $\cdot$ denotes standard complex multiplication. Define $\delta_t = \frac{\phi_t}{\phi_0}$ so that $\varphi_t = \delta_t \cdot \varphi_0$. Define $d_t = \varphi_t/\varphi_0$, and let $a_t = \frac{1}{t}\sum_{i=0}^t d_i$. Using this notation,

$$d_t = \kappa \cdot a_{t-1} \tag{16}$$

$$a_t = ((t-1)a_{t-1} + d_t)/t. \tag{17}$$

We see that $a_t = L_{t,K}(a_{t-1})$ is an affine map with slope $((t-1)+\kappa)/t$ and intercept $0$. Suppose that the characteristic function of the density converges to $\varphi_\infty$. Then the map $a_t$ has a fixed point. As long as $\kappa \neq 1$, this fixed point must satisfy the equation

$$\varphi = ((t-1)+\kappa)\varphi$$
$$\Rightarrow 0 = ((t-1)+\kappa)/g - 1)\,\varphi$$
$$\Rightarrow 0 = (-1+\kappa)\,\varphi \Rightarrow \varphi = 0.$$

Note that if $\varphi_\infty = 0$, its inverse FT is a function that has $0$ probability density everywhere in probability space. Equivalently, the variance of $f_t$ diverges to $\infty$.

$\square$

Although the NLL eventually diverges in the accumulate case, it is clear from the expression for $a_t$ that this divergence occurs very slowly.

For a Gaussian kernel, both the replace and accumulate case offer an interesting shared insight. Throughout the iterative fitting process, regardless of whether we accumulate or replace, the bandwidth monotonically grows. Therefore, when one starts this process with a very small bandwidth smaller than the optimal bandwidth for the density being fit, one could initially observe a decrease in the negative log likelihood as the bandwidth approaches its optimum.

Finally, model collapse, while inevitable with a fixed bandwidth, can be avoided in all cases by shrinking the bandwidth at a sufficiently fast rate. Since practitioners typically optimize their bandwidth according to the amount of the data that they have, the bandwidth should have the form $c(tn)^{1/5}$ where $c$ is a constant. In this setting, model collapse is avoided entirely.

**Theorem 7.** *Suppose that data accumulates as in Section 2.2 for a Gaussian kernel. Let the bandwidth at the $n$th model-fitting iteration be $c(tn)^{-1/5}$ for a constant $c$. Then the asymptotic variance of the limiting KDE is finite.*

*Proof.* Let $K_{c(tn)^{-1/5}}$ denote the kernel at the $t$th model-fitting iteration. Let $f_0$ denote the original distribution, and define $f_t$ to be the distribution of the KDE at the $t$th iteration.

We can write

$$f_t = \frac{1}{t}\sum_{i=1}^t f_{i-1} * K_{c(in)^{-1/5}}$$

$$= \left(1 - \frac{1}{t}\right) \cdot \left(\frac{1}{t-1}\sum_{i=1}^{t-1} f_{i-1} * K_{c(in)^{-1/5}}\right) + \frac{1}{t}f_{t-1} * K_{c(tn)^{-1/5}}$$

$$= \left(1 - \frac{1}{t}\right) f_{t-1} + \frac{1}{t}f_{t-1} * K_{c(tn)^{-1/5}}$$

$$= \left(\left(1 - \frac{1}{t}\right) K_0 + \frac{1}{t}K_{c(tn)^{-1/5}}\right)$$

where $K_0$ is the identity kernel, or equivalently the Gaussian kernel with $0$ bandwidth.

Therefore, we find that

$$f_t = f_0 * \circledast_{i=1}^t \left(\left(1 - \frac{1}{i}\right) K_0 + \frac{1}{i}K_{c(in)^{-1/5}}\right).$$

Define $W_i$ to be a random variable that is $K_{c(in)^{-1/5}}$ with probability $\frac{1}{i}$ and $K_0$ with probability $1 - \frac{1}{i}$. We can rewrite $X_t$, a random variable drawn at the $t$th fitting iteration as

$$X_t = X_0 + \sum_{i=1}^{t} W_i.$$

All of $X_0, W_1, ..., W_t$ are independent. The variance is given by

$$\text{Var}(X_t) = \text{Var}(X_0) + \sum_{i=1}^{t} \text{Var}(W_i)$$

$$= \text{Var}(X_0) + \sum_{i=1}^{t} \frac{1}{i} \times \frac{c}{(in)^{2/5}}$$

$$= \text{Var}(X_0) + \frac{c}{n^{2/5}} \sum_{i=1}^{t} \frac{1}{i^{7/5}}.$$

As $t \to \infty$,

$$\text{Var}(X_t) \to \text{Var}(X_0) + \frac{c}{n^{2/5}} \sum_{i=1}^{\infty} \frac{1}{i^4} < \infty.$$

Therefore, when the kernel size is appropriately adjusted, the variance of the KDE under accumulate converges. $\qquad\square$

## C  EXPERIMENTAL RESULTS: SWEEP CONFIGURATIONS

### C.1  MODEL COLLAPSE IN MULTIVARIATE GAUSSIAN MODELING

To study model collapse in multivariate Gaussian modeling, we ran the following YAML sweep:

```
program: src/fit_gaussians/fit_gaussians.py
entity: rylan
project: rerevisiting-model-collapse-fit-gaussians
method: grid
parameters:
  data_dim:
    values: [ 1, 3, 10, 31, 100 ]
  num_samples_per_iteration:
    values: [10, 32, 100, 316, 1000]
  num_iterations:
    values: [ 100 ]
  seed:
    values: [ 0, 1, 2, 3, 4, 5, 6, 7, 8, 9, 10, 11, 12, 13, 14, 15, 16, 17, 18, 19,
  setting:
    values: [
      "Accumulate",
      "Accumulate-Subsample",
      "Replace",
    ]
  sigma_squared:
    values: [
      1.0,
    ]
```

Seeds were swept from 0 to 99, inclusive.

### C.2  MODEL COLLAPSE IN KERNEL DENSITY ESTIMATION

To study model collapse in multivariate Gaussian modeling, we ran the following YAML sweep:

Seeds were swept from 0 to 99, inclusive.

### C.3  MODEL COLLAPSE IN KERNEL DENSITY ESTIMATION

To study model collapse in kernel density estimation, we ran the following YAML sweep:

```
program: src/fit_kdes/fit_kdes.py
entity: rylan
project: rerevisiting-model-collapse-fit-kdes
method: grid
parameters:
  data_config:
    parameters:
      dataset_name:
        values: ["blobs"]
      dataset_kwargs:
        parameters:
          n_features:
            values: [2]
  kernel:
    values: ["gaussian"]
  kernel_bandwidth:
    values: [0.1, 0.5, 1.0]
```

```
num_samples_per_iteration :
  values : [10 , 32 , 100 , 316 , 1000]
num_iterations :
  values : [ 100 ]
seed :
  values : [ 0, 1, 2, 3, 4, 5, 6, 7, 8, 9, 10, 11, 12, 13, 14, 15, 16, 17, 18, 19,
setting :
  values : [
    "Accumulate" ,
    "Accumulate−Subsample" ,
    "Replace" ,
  ]

program : src / fit_kdes / fit_kdes . py
entity : rylan
project : rerevisiting −model−collapse −fit −kdes
method : grid
parameters :
  data_config :
    parameters :
      dataset_name :
        values : [" circles "]
      dataset_kwargs :
        parameters :
          noise :
            values : [0.05]
  kernel :
    values : [" gaussian "]
  kernel_bandwidth :
    values : [0.1 , 0.5 , 1.0]
  num_samples_per_iteration :
    values : [10 , 32 , 100 , 316 , 1000]
  num_iterations :
    values : [ 100 ]
  seed :
    values : [ 0, 1, 2, 3, 4, 5, 6, 7, 8, 9, 10, 11, 12, 13, 14, 15, 16, 17, 18, 19,
  setting :
    values : [
      "Accumulate" ,
      "Accumulate−Subsample" ,
      "Replace" ,
    ]

program : src / fit_kdes / fit_kdes . py
entity : rylan
project : rerevisiting −model−collapse −fit −kdes
method : grid
parameters :
  data_config :
    parameters :
      dataset_name :
        values : [" moons "]
      dataset_kwargs :
        parameters :
          noise :
            values : [0.05]
  kernel :
    values : [" gaussian "]
  kernel_bandwidth :
```

```
      values: [0.1, 0.5, 1.0]
    num_samples_per_iteration:
      values: [10, 32, 100, 316, 1000]
    num_iterations:
      values: [ 100 ]
    seed:
      values: [ 0, 1, 2, 3, 4, 5, 6, 7, 8, 9, 10, 11, 12, 13, 14, 15, 16, 17, 18, 19, 
    setting:
      values: [
        "Accumulate",
        "Accumulate-Subsample",
        "Replace",
      ]

program: src/fit_kdes/fit_kdes.py
entity: rylan
project: rerevisiting-model-collapse-fit-kdes
method: grid
parameters:
  data_config:
    parameters:
      dataset_name:
        values: ["swiss_roll"]
      dataset_kwargs:
        parameters:
          noise:
            values: [0.05]
    kernel:
      values: ["gaussian"]
    kernel_bandwidth:
      values: [0.1, 0.5, 1.0]
    num_samples_per_iteration:
      values: [10, 32, 100, 316, 1000]
    num_iterations:
      values: [ 100 ]
    seed:
      values: [ 0, 1, 2, 3, 4, 5, 6, 7, 8, 9, 10, 11, 12, 13, 14, 15, 16, 17, 18, 19, 
    setting:
      values: [
        "Accumulate",
        "Accumulate-Subsample",
        "Replace",
      ]
```

Seeds were swept from 0 to 99, inclusive.

C.4   MODEL COLLAPSE IN LINEAR REGRESSION

To study model collapse in linear regression, we ran the following YAML sweep:

```
program: src/fit_linear_regressions/fit_linear_regressions.py
entity: rylan
project: rerevisiting-model-collapse-fit-lin-regr
method: grid
parameters:
  data_dim:
    values: [ 100, 10, 31, 3, 1 ]
  num_samples_per_iteration:
    values: [10, 32, 100, 316, 1000]
```

```
num_iterations:
  values: [ 100 ]
seed:
  values: [ 0, 1, 2, 3, 4, 5, 6, 7, 8, 9, 10, 11, 12, 13, 14, 15, 16, 17, 18, 19, 
setting:
  values: [
    "Accumulate",
    "Accumulate-Subsample",
    "Replace",
  ]
sigma_squared:
  values: [
    0.1, 1.0, 10.
  ]
```

Seeds were swept from 0 to 99, inclusive. Note: We ran this sweep as 9 separate sweeps; to understand why, see this GitHub issue.

# D ADDITIONAL EXPERIMENTAL RESULTS FOR MODEL COLLAPSE HYPERPARAMETERS

Due to space limitations in the main text, we can oftentimes only present a subset of runs corresponding to a subset of hyperparameters. We present additional figures with a wide range of hyperparameters here for completeness.

## D.1 ADDITIONAL RESULTS FOR MODEL COLLAPSE IN LINEAR REGRESSION

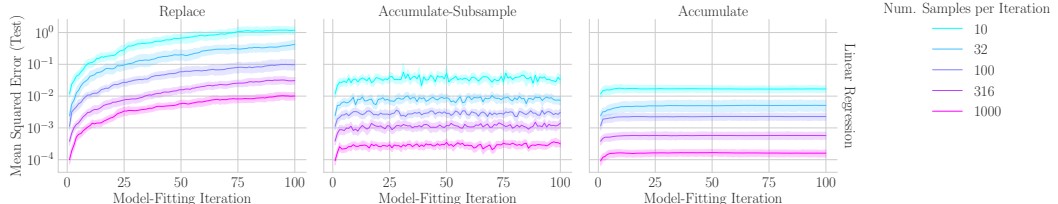

Figure 6: Linear Regression for Data Dimension $d = 1$ and variance $\sigma^2 = 0.10$.

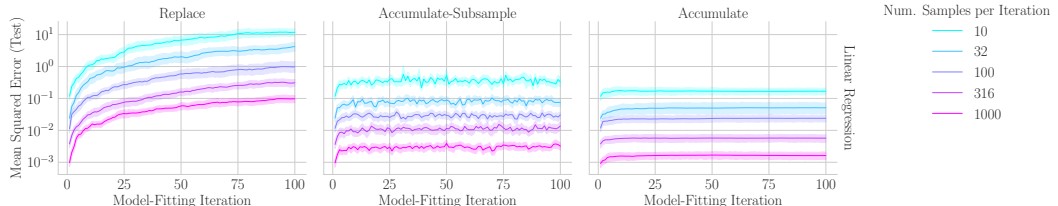

Figure 7: Linear Regression for Data Dimension $d = 1$ and variance $\sigma^2 = 1.00$.

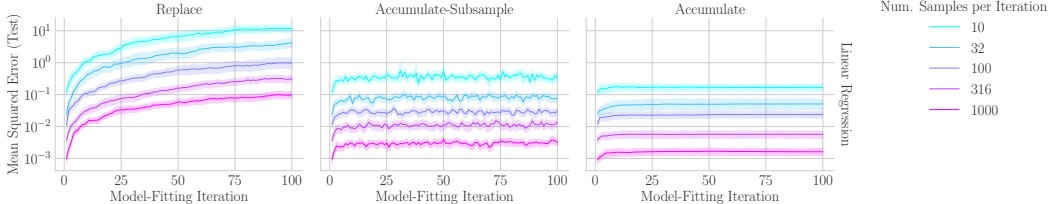

Figure 8: Linear Regression for Data Dimension $d = 1$ and variance $\sigma^2 = 10.0$.

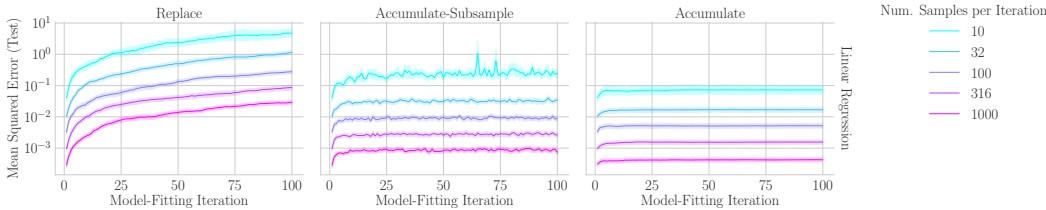

Figure 9: Linear Regression for Data Dimension $d = 3$ and variance $\sigma^2 = 0.10$.

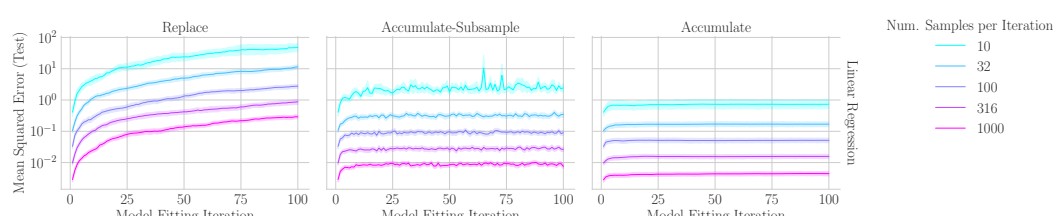

Figure 10: Linear Regression for Data Dimension $d = 3$ and variance $\sigma^2 = 1.00$.

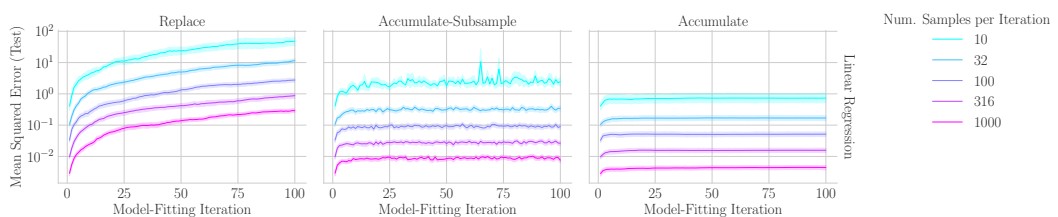

Figure 11: Linear Regression for Data Dimension $d = 3$ and variance $\sigma^2 = 10.0$.

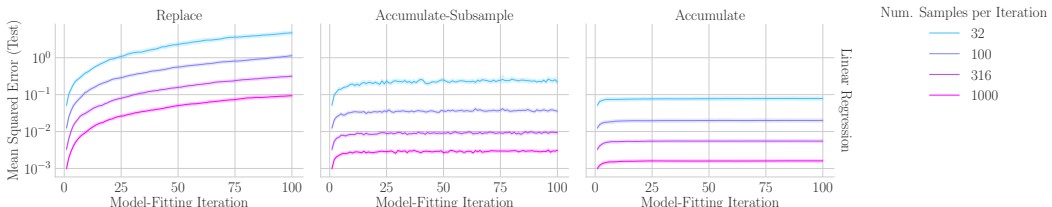

Figure 12: Linear Regression for Data Dimension $d = 10$ and variance $\sigma^2 = 0.10$.

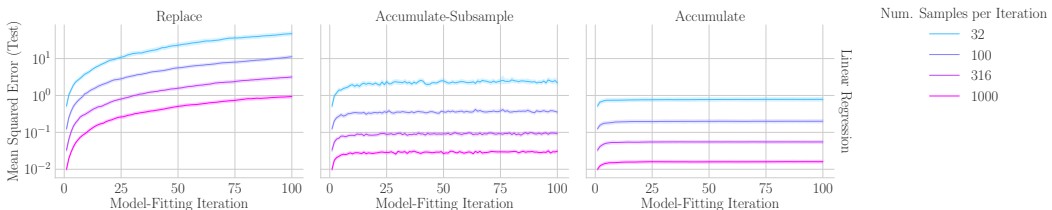

Figure 13: Linear Regression for Data Dimension $d = 10$ and variance $\sigma^2 = 1.00$.

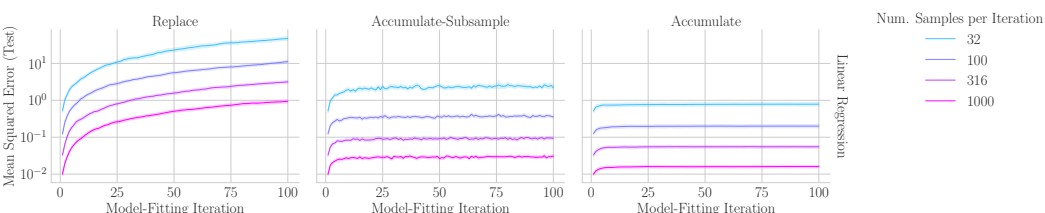

Figure 14: Linear Regression for Data Dimension $d = 10$ and variance $\sigma^2 = 10.0$.

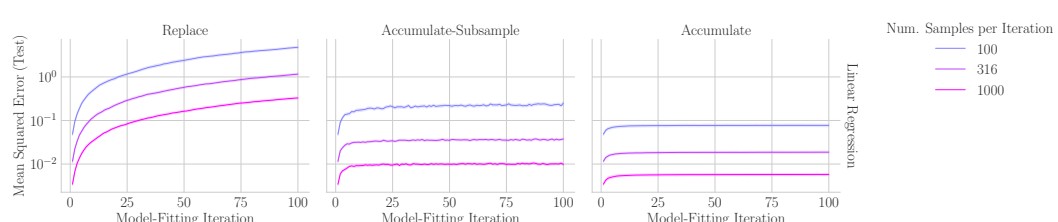

Figure 15: Linear Regression for Data Dimension $d = 32$ and variance $\sigma^2 = 0.10$.

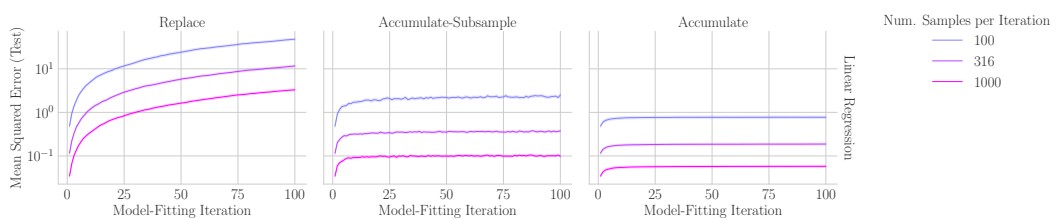

Figure 16: Linear Regression for Data Dimension $d = 32$ and variance $\sigma^2 = 1.00$.

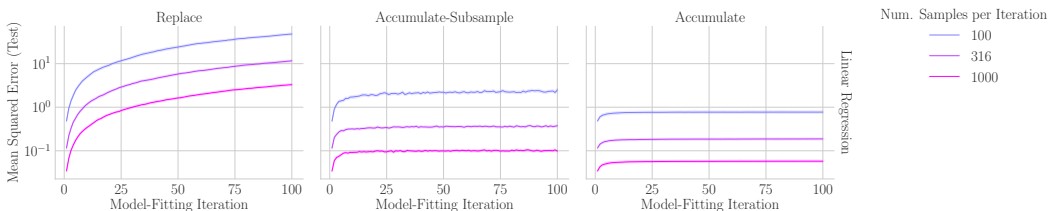

Figure 17: Linear Regression for Data Dimension $d = 32$ and variance $\sigma^2 = 10.0$.

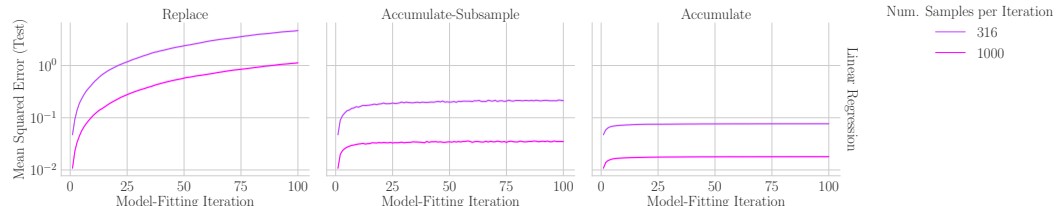

Figure 18: Linear Regression for Data Dimension $d = 100$ and variance $\sigma^2 = 0.10$.

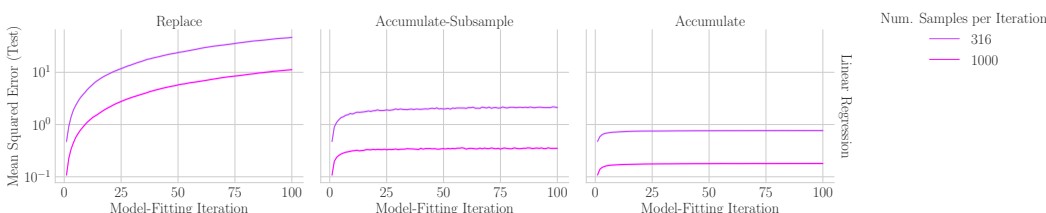

Figure 19: Linear Regression for Data Dimension $d = 100$ and variance $\sigma^2 = 1.00$.

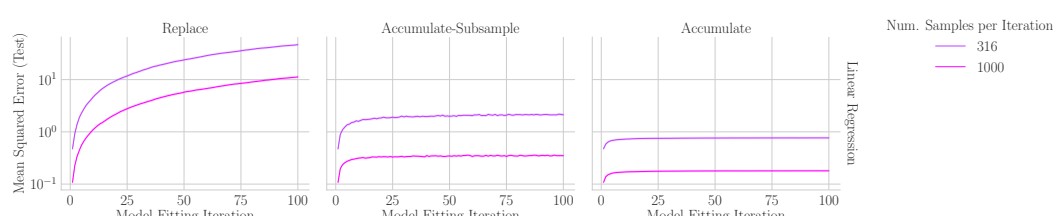

Figure 20: Linear Regression for Data Dimension $d = 100$ and variance $\sigma^2 = 10.0$.

