# OpenReview forum: "Collapse or Thrive? Perils and Promises of Synthetic Data in a Self-Generating World"
_ICLR.cc/2025/Conference — Submitted to ICLR 2025_

### Official Review · Reviewer_9sJL · 2024-11-03

**Soundness:** 2
**Presentation:** 3
**Contribution:** 3
**Rating:** 6
**Confidence:** 3

**Summary:**

This paper addresses the risks of training generative models on datasets increasingly dominated by AI-generated (synthetic) data, studying "model collapse"—where model performance deteriorates as synthetic data accumulates. Through theoretical and empirical analyses, the authors explore when and how this collapse can be mitigated. They test three data-handling approaches (replace, accumulate, and accumulate-subsample) across various generative model settings. Findings suggest that model collapse is avoidable if synthetic data is accumulated alongside real data, and highlight how different ratios of real to synthetic data influence model performance.

**Strengths:**

1. The studied problem of model collapse under increasing AI-generatd data is prompt and interesting.

2. The theories and empirical studies round up the good work.

3. This paper offers actionable guidance on how to balance real and synthetic data in training.

**Weaknesses:**

1. Some of the theoretical proofs hinge on idealized assumptions (e.g., independence and Gaussian distribution in certain proofs). These may not fully apply in practical, real-world datasets.

2. The "accumulate-subsample" paradigm offers a practical perspective on fixed compute budgets but is tested under simplified conditions. Additional real-world constraints (e.g., dynamic memory handling) are not considered in depth.

3. While language models are likely the most relevant generative model for readers, the theoretical analysis does not extend to this model type and is instead focused on Gaussian and kernel density estimation models. Additionally, the size of the language model used in this study is quite small. While contemporary models like Llama range from 7B to 70B parameters, the paper’s experiments are limited to Gemma2 with only 2B parameters.

**Questions:**

1. The study's empirical analysis is limited to a 2B parameter language model. Given the prevalence of larger models like Llama 3 and Mistral, do the authors plan to extend their experiments to these more widely used models to assess the generalizability of their findings?

2. This work primarily explores synthetic data quantity, but the quality of synthetic data is also a significant factor. Would lower-quality synthetic data exacerbate collapse, and could filtering or curating synthetic data mitigate this effect?

---

> ### Author Response · Authors · 2024-11-14
> **Rebuttal to Reviewer 9sJL**
>
> Thank you for reviewing our paper. We are grateful you found the problem of model collapse interesting, our theoretical and empirical results good and our guidance actionable.
>
> For calibration purposes, we’d like to note that the ICLR 2025 rubric differs slightly from previous similar conferences. For example:
>
> - To indicate "Accept", the NeurIPS 2024 rubric says to use 7 whereas the ICLR 2025 rubric says to use 8
> - To indicate "Strong Accept", the NeurIPS 2024 rubric says to use 9 whereas the ICLR 2025 rubric says to use 10
>
>
> To address the concerns you raised:
>
> > Some of the theoretical proofs hinge on idealized assumptions (e.g., independence and Gaussian distribution in certain proofs). These may not fully apply in practical, real-world datasets.
>
> We examine Gaussian models because these were a canonical example of how models collapse provided in Shumailov et. al, a Nature publication with ~250 citations.  Although the assumptions do not extend to training deep models, the patterns observed in multivariate Gaussian modeling were consistent across kernel density estimation, supervised finetuning of language models, linear regression and pretraining of language models.  Thus, Gaussian models and KDEs appear to qualitatively match generative models that are too difficult to study mathematically.
>
> >The "accumulate-subsample" paradigm offers a practical perspective on fixed compute budgets but is tested under simplified conditions. Additional real-world constraints (e.g., dynamic memory handling) are not considered in depth.
>
> While there are many settings we could study, we intentionally chose to study fixed compute budgets because this setting has significant real world implications. While accumulating data avoids model collapse, training on all accumulating data rapidly increases the cost of model training. Thus, it is highly unlikely that future large-scale deep generative models will be trained on 10 quadrillion or 100 quadrillion tokens; rather, it is far more likely that web-scale datasets will be collected and filtered to a smaller size. Section 3 and Figure 4 is a careful examination of 5 different generative modeling settings to study this crucial middle ground.
>
> If you feel other settings have important real world ramifications, please let us know. We would very much appreciate good suggestions!
>
> > the size of the language model used in this study is quite small. While contemporary models like Llama range from 7B to 70B parameters, the paper’s experiments are limited to Gemma2 with only 2B parameters.
>
> What experiment(s) would you most like to see? We are happy to run additional experiments, but our compute budget is finite and the review timeline is finite, so we want to make sure we prioritize the experiments you think would be most valuable. For instance, we can extend the SFT results (Figure 3) to include other language models, e.g., Gemma 2 9B and Gemma 2 27B. What would you advise?
>
> >This work primarily explores synthetic data quantity, but the quality of synthetic data is also a significant factor. Would lower-quality synthetic data exacerbate collapse, and could filtering or curating synthetic data mitigate this effect?
>
> We agree that the quality of synthetic data is likely to play a significant role. We view our results as an (approximate) upper bound on the harm of synthetic data since we want to understand the effects of unfiltered data of arbitrary quality inundating future pretraining data. If data filtering is applied, we imagine that performance would only lead to better models.

---

> ### Author Response · Authors · 2024-11-24
> **Rebuttal to Reviewer 9sJL (Part 2)**
>
> In accordance with your suggestion that experiments on larger models would be more realistic, we have started SFT experiments on Gemma-2-9B and Gemma-2-27B (see figure 3).  The preliminary findings from the first several training iterations are displayed in Figure 3 and are consistent with our findings for Gemma-2-2B.
>
> Additionally, we added a Related Works section to the updated manuscript.
>
> In light of these new experiments and paper improvements, we hope that you will consider increasing your score.

---

> ### Author Response · Authors · 2024-12-03
> **Final Reminder to 9sJL**
>
> Dear 9sJL,
>
> We expended considerable compute and effort to run experiments on Gemma 9B and 27B.  We ran these experiments to address your concerns that our models might not be large enough to provide a realistic picture of what could happen in model collapse.  We measured results that were consistent with our findings on smaller models.  Despite our efforts, we have not received any follow-up from you.  Given that the review period is ending, we want to take this final opportunity to urge you to assess our significantly improved manuscript and update your score if you think that we have addressed your concerns.
>
> Thank you,
>
> The Authors

---

### Official Review · Reviewer_BKEn · 2024-11-03

**Soundness:** 4
**Presentation:** 2
**Contribution:** 2
**Rating:** 3
**Confidence:** 3

**Summary:**

The paper investigates the impact of iteratively training a generative model on a combination of real and synthetic data generated from a previously trained model. This is an important area of research to explore and understand, as exclusively training models on synthetic data will eventually fail, as highlighted by prior work.
The authors conduct experiments to validate previous findings, offering new evidence to clarify cases in which model collapse can be avoided. They also explore additional experimental settings in which they control for the ratio of synthetic/real data and the absolute quantity of real data to gain further insight into the limits of training with synthetic data.

**Strengths:**

1. Motivation. The research question is clearly defined and contextually relevant. Exploring the impact of training with synthetically generated data is important for advancing the field of machine learning.

2. Experiment section. The experiments are thorough, relevant, and insightful. The authors verify that the hypotheses they are testing hold across a wide range of models and data generation processes. They present interesting experiments that control for different parameters, offering clear insights into the scope of the problem. The authors clearly list the questions they aim to answer and provide solid experiments to support their conclusions, along with clear and detailed explanations of their experimental setup.

3. Writing and structure. The paper is well written and follows a logical and clear structure.

**Weaknesses:**

1. Contribution. The majority of the experimental section (4/9 pages) validates the findings of previous work (Gerstgrasser et al., 2024). The experiments verify that retraining with a mix of synthetic and real data can avoid model collapse in three additional settings. Although it is valuable to reproduce and verify previous findings, this should not constitute the main contribution of a novel work. Emphasizing and clarifying the unique contributions and findings of this paper would strengthen its impact.

2. Presentation. Overall, the presentation of the work could be improved in several ways. First, a clearer discussion of related work to better position this paper's contributions is needed. In the introduction, various papers on model collapse are listed with the comment that they have differing methodologies and conclusions. What are the differing methodologies and conclusions? A brief summary of these differences and a clearer statement of this work's position relative to others is required. This should be easy to achieve as the paper is one page short of the limit, so I am not sure what justifies the almost nonexistent discussion of the listed previous work. Second, the presentation of figures could be improved (e.g., readability of axis labels, font size, relative scaling of the plots). Some figures use notation like 1.2×10^0 instead of simply 1.2, and there is a lot of wasted empty space in the figures. Lastly, there are several errors in the references, as several published works are cited as arXiv references. This gives the impression of an unpolished work.

Minor

1. Some of the writing is needlessly sensationalized. For example, describing "that model collapse is caused by deleting past data en masse and avoided by instead accumulating real and synthetic data" as a "provocative pair of claims" feels excessive. It’s clear that using real data works and that purely generated data does not, so it is not surprising that a middle ground exists. Especially since this finding was already made by previous work.

2. The use of tweets and LinkedIn posts as references gives the paper an unserious tone, though this may be a matter of personal preference.

**Questions:**

1. This paper presents iterative settings in which performance either suffers from model collapse (replace) or avoids it (accumulate and accumulate+subsample). Have the authors explored ways to identify the breaking point between these two modes? For example, would the results hold if, instead of adding $n$ samples at each iteration, we add an increasing amount of synthetic data (say, $2n$ each time)?

2. Another question is about the relative quality of the synthetic data. Presumably, there is a connection between how well the model learns the task and whether its generated data can effectively train a second iteration. Do the authors have comments on this?


3. Regarding the observations in the blob experiment, the authors note a tendency: "Interestingly, for specific pairs of datasets and numbers of samples per iteration, training on real data while accumulating synthetic data can yield lower loss on real test data than training on real data alone." For these specific settings where the true distribution is known (I believe that it is the case?), it would be valuable to report the NLL value of the true distribution as a reference point.

4. Have the authors considered extending the result from Theorem 1 to the accumulate-subsample setup? Or relating this result to their experimental results? This could strengthen the contribution of this work.

---

> ### Author Response · Authors · 2024-11-14
> **Rebuttal to Reviewer BKEn (part 1)**
>
> Thank you for reviewing our paper. We are grateful you found the motivation clear and relevant and the experiments thorough and insightful.
>
> For calibration purposes, we’d like to note that the ICLR 2025 rubric differs slightly from previous similar conferences. For example:
>
> - To indicate "Accept", the NeurIPS 2024 rubric says to use 7 whereas the ICLR 2025 rubric says to use 8
> - To indicate "Strong Accept", the NeurIPS 2024 rubric says to use 9 whereas the ICLR 2025 rubric says to use 10
>
>
> To address the concerns you raised:
>
> > Some of the writing is needlessly sensationalized. For example, describing "that model collapse is caused by deleting past data en masse and avoided by instead accumulating real and synthetic data" as a "provocative pair of claims" feels excessive.
>
> We concur. The final version will not have this sentence, and we have removed other sensationalist language. We will post a revised manuscript soon.
>
> > The use of tweets and LinkedIn posts as references gives the paper an unserious tone
>
> Agreed. Our goal was to communicate concerns with model collapse (or unfiltered synthetic data) from industry leaders steering billions of dollars who do not write academic papers, but the footnotes indeed felt unserious. We have replaced them with proper citations. Does this seem like an appropriate solution?
>
> > Contribution. The majority of the experimental section (4/9 pages) validates the findings of previous work (Gerstgrasser et al., 2024) [...] Although it is valuable to reproduce and verify previous findings, this should not constitute the main contribution of a novel work.
>
> Hopefully we can clarify why we feel this manuscript contains multiple novel insights.
>
> 1.  The first third of our manuscript tests whether the data paradigm claims of Gerstgrasser hold in the model settings of Shumailov. In our view, this is not a reproduction; rather, we are testing whether a prediction holds in new settings. Such work is a fundamental component of science and should not be dismissed as lacking novelty.
> 2.  The experiments in the first third of our manuscript significantly extend beyond either paper. Gerstgrasser considers the settings of language model pretraining, linear regression, VAEs, and diffusion. In contrast, we consider Gaussian mean/covariance estimation, kernel density estimation, language model instruction finetuning.  Moreover, Gerstgrasser experiments on TinyStories only ran 5 model-fitting iterations, whereas we go to over 40 model-fitting iterations for Replace and to 10+ iterations for Accumulate.
> 3.  The second third of our manuscript introduces a middle ground for how data are treated between Accumulate and Replace. This data paradigm might be conceptually simple, but it has significant real world implications and thus needs to be studied carefully. While accumulating data avoids model collapse, training on all accumulating data rapidly increases the cost of model training. Thus, it is highly unlikely that large-scale deep generative models will be trained on 10 quadrillion or 100 quadrillion tokens, and is instead far more likely that web-scale datasets will be collected and filtered to a smaller size. Section 3 and Figure 4 is a careful examination of 5 different generative modeling settings to study this crucial middle ground.
> 4.  The final third of our manuscript moves far beyond prior work to tease apart the relative importance of proportionality versus cardinality of real and synthetic data on SFT performance, which previous papers on model collapse did not directly compare, but turns out to be crucial. We also provide insights into the value of synthetic data, discovering that when the number of real data is insufficient, a specific amount of synthetic data can improve model performance.
>
> We will rewrite the manuscript to clarify these contributions.
>
> > A clearer discussion of related work to better position this paper's contributions is needed [...] A brief summary of these differences and a clearer statement of this work's position relative to others is required.
>
> This is a good suggestion for improving the manuscript. We will add a Related Work section.
>
> > The presentation of figures could be improved (e.g., readability of axis labels, font size, relative scaling of the plots). Some figures use notation like 1.2×10^0 instead of simply 1.2, and there is a lot of wasted empty space in the figures
>
> We will update the figures with additional data and work to improve them.
>
> > Lastly, there are several errors in the references, as several published works are cited as arXiv references. This gives the impression of an unpolished work.
>
> Thank you for your keen eye. Could you please tell us which papers caught your attention? We will update whichever papers we have outdated citations for.

---

> ### Author Response · Authors · 2024-11-14
> **Rebuttal to Reviewer BKEn (part 2)**
>
> > Have the authors explored ways to identify the breaking point between these two modes?
>
> The second third of our paper (Accumulate-Subsample) is one way to explore the boundary between Replace and Accumulate, as is the final third of our paper (cardinality versus proportionality of synthetic data).  Theoretically, the Gaussian case and KDE settings suggest that as long as one does not increase the amount of synthetic data added at each iteration superlinearly, collapse does not occur.  We are happy to emphasize these points in the final version of the paper.
>
> If there is a specific analysis that you think would be insightful, please let us know.
>
> > Another question is about the relative quality of the synthetic data. Presumably, there is a connection between how well the model learns the task and whether its generated data can effectively train a second iteration. Do the authors have comments on this?
>
> We think this is a fascinating question that we intend to explore in a follow-up paper. To briefly sketch the methodology, we select model families with different scaling dimensions (parameters, data, compute), then sample and continue pretraining using a mixture of real and synthetic data. The central research question would be aimed at relating how well the first iteration model has learned the task to what effect the first model’s synthetic data has on the second iteration model.
>
> > For these specific settings where the true distribution is known (I believe that it is the case?), it would be valuable to report the NLL value of the true distribution as a reference point.
>
> We will add these figures to the appendix, along with other additional figures.  For some of the KDE experiments, the NLL with respect to the true distribution is quite difficult to calculate.
>
> > Have the authors considered extending the result from Theorem 1 to the accumulate-subsample setup? Or relating this result to their experimental results? This could strengthen the contribution of this work.
>
> This would be a valuable extension but we were unable to derive a result. If you see a path to doing so, please let us know.

---

> ### Author Response · Authors · 2024-11-24
> **Rebuttal to Reviewer BKEn (part 3)**
>
> In accordance with your advice, we have made the following changes to the updated manuscript:
> - We included a Related Works section.  We are open to any suggestions on works that we missed.
> - We removed the $\times 10^0$ from the axes of the plots.
> - We removed colloquial and sensational language from the manuscript.
> - We updated pre-print citations with the proceedings citations.  We changed the following citations:  Alemohammad (2024), Bertrand (2023), Briesch et al. (2023), Brown et al. (2020), Dohmatob (2024a, 2024b), Gerstgrasser et al. (2024), Guo et al. (2023), Martinez et al. (2023), Padmakumar (2024), Penedo (2024), Sachdeva (2024), Shumailov (2023), Wettig (2024).  Please let us know if you see others that are outdated.
>
> Additionally, we have run several experiments on Gemma-2-9B and Gemma-2-27B to show consistency of our SFT findings across larger models and to address your question about how the quality of the synthetic data affects model collapse.  The results from the first several training iterations can be found in Figure 3.
>
> Thank you again for these helpful suggestions.  We hope that in light of these improvements, you will raise your score.

---

> > ### Author Response · Authors · 2024-11-27
> > **Request for feedback from Reviewer BKEn**
> >
> > Thank you for your suggestions on our paper.  We have incorporated your feedback to improve our work.
> >
> > Given that today is our final opportunity to update the manuscript, we would appreciate any guidance on whether our rebuttal, revised manuscript, and added experiments have addressed some of your concerns.  If so, we hope that you will consider recalibrating your score to the improved paper.  If not, please let us know what else we can do to make our work better.

---

> > > ### Comment · Reviewer_BKEn · 2024-11-29
> > >
> > > Thank you for you response, and for updating a revised version with the changes.
> > >
> > >
> > > **Contribution.** My issue with novelty has not been addressed by this response. I already acknowledged and agree with the statement that (paraphrasing): ``testing whether a prediction holds in new settings does not necessarily lack novelty and is a fundamental component of science."
> > > However, if you are building on findings from previous work and are claiming novelty, you need to justify why the settings you are investigating are noteworthy.
> > >
> > > I can think of 3 ways that this can be done; 1) The existing settings explored are insufficient in scope. -> I found this point hard to justify; the combined existing experiments in the literature are pretty extensive. 2) The proposed three new settings have a particularity that would make us believe the previous findings might not hold. -> I don't find that this is argued anywhere in the work, nor do I think there is anything that would make us believe that.
> > > 3) The newly explored settings have a particular practical usage that makes empirical results from those settings notable. -> I also don't think this can be argued, as two out of the three settings (Gaussian and KDE) explored in this work would not realistically be used in scenarios where sampling from synthetic data would be done.
> > >
> > >
> > > Therefore, I don't think the claim from point 2:"The experiments in the first third of our manuscript significantly extend beyond either paper" is strongly supported. Not reproducing the exact experiment conducted in previous work does not constitute "significantly extending" the experiments contained in the existing literature, especially if it was already pretty extensive.
> > >
> > > If the authors can provide an alternative reasoning that I missed, I would be happy to reconsider.
> > >
> > > On point 3: Not accumulating data has already been explored in Martínez et al. (2023), Mobahi et al. (2020), and Dohmatob et al. (2024a), as the authors mention in their related work. Then novelty of this second part of the work is therefore not clear to me. A better contextualization of this part in the related work section could clarify that point.
> > >
> > > On point 4: I agree that the last part of the work is more novel and interesting. However, my argument concerns proportionality and the focus of the work. The part I estimate to have very limited novelty constitutes the majority of the paper.
> > > Which is why, taken in its totality, the novelty of this work, in my view, is insufficient.

---

> > > > ### Comment · Reviewer_BKEn · 2024-11-29
> > > >
> > > > **Related Work.** Thank you for adding a related work section. While this might be considered a major revision, I will leave it to the meta-reviewer to decide and will take it as is.
> > > >
> > > > I have significant concerns about the related work section of this manuscript. In my initial review, I stated that proper contextualization of the work was missing. After reviewing the cited literature and the added related work section, I found that it is still the case.
> > > > I will outline my reasoning in the detailed points below.
> > > >
> > > > 1. Contextualizing the work that has been done with respect to assessing the mix of real and synthetic data required to avoid model collapse is crucial. Since this is the main focus of this work, there should be a much more detailed account of the existing literature on this topic. For example, the authors do not mention in this section that Bertrand (2024) directly addresses this question and groups this citation with work that observed model collapse phenomenon (see line 475).
> > > > However, the abstract of Bertrand et al. (2024) explicitly states:
> > > > "In this paper, we develop a framework to rigorously study the impact of training generative models on mixed datasets—from classical training on real data to self-consuming generative models trained on purely synthetic data. We first prove the stability of iterative training under the condition that the initial generative models approximate the data distribution well enough and the proportion of clean training data (w.r.t. synthetic data) is large enough.". This is extremely close to what this paper investigates, but is not acknowledged as such. Moreover, Theorem 1 from Bertrand et al. (2024) has a direct connection to the results presented in this work. In particular:
> > > > ''Using our theoretical framework, we show the stability of iterative retraining of deep generative models by proving the existence of a fixed point (Theorem 1) under the following conditions: (1) The first iteration generative model is sufficiently 'well-trained' and (2) Each retraining iteration keeps a high enough proportion of the original clean data. Moreover, we provide theoretical (Proposition 1) and empirical (Figure 1) evidence that failure to satisfy these conditions can lead to model collapse (i.e., iterative retraining leads the generative model to collapse to outputting a single point)." This result provides the explicit connection between the amount of real and synthetic data and the quality of the synthetic model required to ensure the stability of the iterative retraining process. Theorem 1 from the proposed paper seems to be a special case of this much more general result. Can the authors comment on the relationship between their result and the result from Bertrand et al. (2024)?
> > > >
> > > > 2.  In line 487, the authors introduces papers with the comment: ''A common feature of many of these is that at least some real data is discarded, often because total dataset size is kept constant across model-fitting iterations''. Doesn't this correspond to the setting you are considering in Section 3: (MODEL COLLAPSE UNDER A FIXED COMPUTE BUDGET) ? If that is the case, can the authors comment on how Section 3 form Dohmatob et al. (2024a)) , in particular ''1) In the case of mixing $\pi$-fraction of the original data with a (1-$\pi$) fraction of AI-generated
> > > > data'' for example relates to Section 3 of their paper? Contextualizing Section 3 w.r.t those existing works is important to justify the 2nd contribution of this work.
> > > >
> > > >
> > > >
> > > >
> > > > **Minor Points**
> > > >
> > > > **Tweets**. No I don't think that listing tweets and linkedin posts as proper references is appropriate. If you are to keep them, leave them as footnotes.
> > > >
> > > > **Citations.** Thank you for modifying these. The fifth reference is an ICLR paper.
> > > >
> > > > Typo;
> > > > 13)`` Some authors prophesy''

---

> ### Author Response · Authors · 2024-11-30
> **Response to Reviewer BKEn (Part I)**
>
> Thank you for getting back to us!  We’re excited to engage with you more on the issues that you have raised, and we appreciate the specificity that you have provided in your response.
> >the authors do not mention in this section that Bertrand (2024) directly addresses this question[...]
>
> >Tweets. No I don't think that listing tweets and linkedin posts as proper references is appropriate
>
> >Citations. Thank you for modifying these. The fifth reference is an ICLR paper.
>
> We admire Bertrand’s work, and we’ll expand our discussion of it later in the rebuttal and in the final version of the paper, since we can no longer edit the rebuttal draft.  We will replace the Twitter and LinkedIn references with something more suitable, and we will fix the citation that you flagged.  We hope that these easily-remedied objections to our paper will not remain sticking points for you when you reconsider our score.
>
> You initially gave us a score of 3 on the basis of (1) limited novelty, (2) colloquial language, (3) incorrect citations, (4) Twitter/LinkedIn as sources, and (5) a missing related works section.  We have remedied (2), (3), and (5).  We have agreed to fix (4).  Below, we address (1) in depth.  Regardless of (1), given that we addressed all of your other initial concerns, we hope that you will raise your score on the basis that our work has improved with respect to your criticisms.
>
>
> Now, let’s discuss the novelty of the paper in depth:
>
> ## Novelty of the paper
> We agree with your criteria for novelty.  We assert that our paper meets all three of these criteria:
>
> >  The existing settings explored are insufficient in scope. -> I found this point hard to justify; the combined existing experiments in the literature are pretty extensive.
>
> Unlike extensive past experiments on model collapse, we are asking: does model collapse happen when data is replaced and not happen when data accumulates?  To answer this question for the accumulate paradigm, no directly applicable experiments have been done outside of Gerstgrasser et. al. (2024) and Martinez et. al. (2023) (to our knowledge).  Moreover, Martinez and Gerstgrasser __show opposite results__: Martinez finds catastrophic image degradation under the accumulate paradigm whereas Gerstgrasser does not.  While there are a number of published experiments about real-synthetic data mixtures in a single iteration, our construction with multiple model iterations is sufficiently different that the insights do not transfer.  Note that Bertrand et. al. (2024) ensure that the real data comprises no less than a fixed percentage of the total data pool, while we allow the real data to shrink to an arbitrarily small fraction of the total training data.
>
> > The proposed three new settings have a particularity that would make us believe the previous findings might not hold. -> I don't find that this is argued anywhere in the work, nor do I think there is anything that would make us believe that.
>
> Outside of Gerstgrasser et. al. and Martinez et. al., previous findings on model collapse did not examine the accumulate paradigm seen in our paper, and very little theory existed for this regime.  It was non-obvious to us that we could accumulate data and avoid collapse for Gaussian fitting or KDE’s– we only discovered this by writing out the proofs, which were non-trivial.  If you look closely at the KDE proofs, you’ll find that the story is a little bit more complicated, and collapse can still occur in the accumulate setting if you don’t adjust the bandwidth properly.  Details like this would not have been apparent before writing this paper.  Similarly, we were not sure whether the insights of Gerstgrasser et. al. would extend to SFT.  We had to do the experiments to find out.
>
> >The newly explored settings have a particular practical usage that makes empirical results from those settings notable. -> I also don't think this can be argued, as two out of the three settings (Gaussian and KDE) explored in this work would not realistically be used in scenarios where sampling from synthetic data would be done.
>
> We agree that the KDE and Gaussian cases were a theoretical sanity check.  However, SFT is probably the training phase most conducive to synthetic data in the near term.  There are many specialized tasks for which we don’t have sufficient SFT data, so mixing synthetic data with small amounts of real data is a likely approach for practitioners. Moreover, sites like Reddit provided a wellspring of helpfulness training data, but now that we have AI’s trained to answer questions helpfully, it is likely that their responses could end up in the data mix.

---

> ### Author Response · Authors · 2024-11-30
> **Response to Reviewer BKEn (Part II)**
>
> > Not reproducing the exact experiment conducted in previous work does not constitute "significantly extending" the experiments contained in the existing literature, especially if it was already pretty extensive.
>
> For the accumulate paradigm, very few experiments were done.  Before writing this paper, we were unconvinced that accumulating data would provide a general solution to mitigating model collapse.  While model collapse and data mixtures have been extensively explored, the setting from this work has not.
>
> > Not accumulating data has already been explored in Martínez et al. (2023), Mobahi et al. (2020), and Dohmatob et al. (2024a), as the authors mention in their related work. Then novelty of this second part of the work is therefore not clear to me. A better contextualization of this part in the related work section could clarify that point.
>
> We will add this discussion to the Related Work section.  Let’s outline how these papers differ from the setting that we discussed:
>
> __Martinez et. al. (2024):__ Based on our understanding, Martinez actually considered something closer to the accumulate setting than the accumulate-subsample setting.  However, as Martinez states, “the initial data set is small and the diffusion model is simple.”  Experiments were only run for a single, small model that was unable to produce convincing data even at iteration one.
>
> __Mobahi et. al.__: They consider self-distillation, which is functionally different than the accumulate-subsample case (and closer to replace).  Their work is mostly theoretical and their experiments are much smaller scale than ours.
>
> __Dohmatob et. al. (2024a)__: Again, they consider distillation, which is different from accumulate-subsample, with much smaller models and a heavy emphasis on theory.  The only experiments in this paper are on simulated data and KRR with MNIST.
>
> While these studies present important results in the model collapse literature, none of them cover the accumulate-subsample setting, and none of them use language models for their experiments. Thus, we don’t think that these works present an argument against the novelty of our setting.  However, we should have and will include a better discussion of them in the final version of the paper.

---

> ### Author Response · Authors · 2024-11-30
> **Response to Reviewer BKEn (Part III)**
>
> ## The related work section
> We again don’t want to make this a sticking point– we appreciate any guidance on how to better contextualize our work.  We agree that we should have engaged more with Bertram et. al. in our related work.  Let’s discuss how this paper relates to ours:
>
> >This is extremely close to what this paper investigates, but is not acknowledged as such.
>
> We agree that Bertram et. al. study a similar question, but there are important differences. Note that Bertram et. al. does not use the same setup that we do.  They mix the original data with the generations of the latest model iteration, which is different from mixing the original data with all past model iterations.  They control the fraction of real data, whereas we decrease it every iteration.
>
> >Moreover, Theorem 1 from Bertrand et al. (2024) has a direct connection to the results presented in this work. In particular: ''Using our theoretical framework, we show the stability of iterative retraining of deep generative models by proving the existence of a fixed point (Theorem 1) under the following conditions: (1) The first iteration generative model is sufficiently 'well-trained' and (2) Each retraining iteration keeps a high enough proportion of the original clean data
>
> This is a good theorem, and perhaps we can adapt it to our setting!  However, as stated in Bertram et. al., Theorem 1 doesn’t apply to our setting because we drop assumption (2) and let the fraction of real data go to 0.
>
> > Each retraining iteration keeps a high enough proportion of the original clean data. Moreover, we provide theoretical (Proposition 1) and empirical (Figure 1) evidence that failure to satisfy these conditions can lead to model collapse (i.e., iterative retraining leads the generative model to collapse to outputting a single point)."
>
> We actually show something quite different.  We show empirically and in theoretical examples that even when we drop assumption 2, our models do not always collapse.
>
> > This result provides the explicit connection between the amount of real and synthetic data and the quality of the synthetic model required to ensure the stability of the iterative retraining process. Theorem 1 from the proposed paper seems to be a special case of this much more general result.
>
> Our Theorem 1 is not a special case of Bertram et. al.’s Theorem 1.  Bertram et. al. train on the original data and the data from the latest model.  We train on all iterations.
>
> >Can the authors comment on the relationship between their result and the result from Bertrand et al. (2024)?
>
> Bertram’s paper asks a similar question to ours in a different way.  They ask: if the training data contains a guaranteed minimum amount of real data, do models collapse?  We ask: if we let the fraction of real data shrink to 0, but we keep the number of training data points constant, do models collapse?  In both cases, the answer is apparently no.  This fits into our question of whether the proportion or absolute amount of real data is more important when preventing collapse.  We’ll mention this in our revision.
>
> Another salient difference between our paper and Bertram et. al. is that they focus on image models, while we focus on language models.
>
> >A common feature of many of these is that at least some real data is discarded, often because total dataset size is kept constant across model-fitting iterations''. Doesn't this correspond to the setting you are considering in Section 3: (MODEL COLLAPSE UNDER A FIXED COMPUTE BUDGET) ?
>
> There are many different ways in which real data can be discarded.  In Section 3, we sample from the pool of real and synthetic data, probabilistically allowing the fraction of real data in our training set to go to 0.  This is different from what Dohmatob et. al. (2024b) do because they maintain a fixed fraction of real data and study scaling laws in the number of data points.  We’ll contextualize our work better in a revised related works section.
>
> ## Minor grammatical points
>
> >Typo; 13)`` Some authors prophesy''
>
> This actually is correct English (https://www.merriam-webster.com/dictionary/prophesy).  Prophesy is a present tense pl. verb.  However, we can change this if it hurts our score.

---

> ### Author Response · Authors · 2024-12-03
> **Final Reminder to BKEn**
>
> Dear BKEn,
>
> In your last response, you stated that “If the authors can provide an alternative reasoning that I missed, I would be happy to reconsider.”  Given that the review period is ending tomorrow, we wanted to issue a final reminder: we hope that you will consider the arguments that we presented.  We believe that we addressed your concerns thoroughly with alternative reasoning in our latest response, and I hope you will update your score if you agree.
>
> Thank you,
>
> The Authors

---

### Official Review · Reviewer_xzu3 · 2024-11-04

**Soundness:** 3
**Presentation:** 3
**Contribution:** 3
**Rating:** 6
**Confidence:** 3

**Summary:**

this work study the model collapse with the sythetic data.

**Strengths:**

1. The research topic is very interested. It tries to study the model collapse issue when the world have more and more sythetic data due to generative model.

2. The paper contain a lot of experiemts to valdiate their claim.

3. The finding in this work is interesting. they find a non-trivial interaction between real and synthetic data, where the value of synthetic
data for reducing model test loss depends on the absolute quantity of real data.

**Weaknesses:**

1. the paper starts with MULTIVARIATE GAUSSIAN MODELING, can the findings still hold for other model, like VAE, diffusion model, or Transformer. Somehow,  I feel the paper make too strong claim since there are many generative model.

2. the reprodubility is unclear at this time.

3. the math proof in Appendix is mainly for Guassina model, it is unclear how it can generalize to more generative models.

**Questions:**

1. the paper starts with MULTIVARIATE GAUSSIAN MODELING, can the findings still hold for other model, like VAE, diffusion model, or Transformer. Somehow,  I feel the paper make too strong claim since there are many generative model.

2. the reprodubility is unclear at this time.

3. the math proof in Appendix is mainly for Guassina model, it is unclear how it can generalize to more generative models.

---

> ### Author Response · Authors · 2024-11-14
> **Rebuttal to Reviewer xzu3**
>
> Thank you for reviewing our paper. We are excited that you also find this research topic and our findings very interesting, and that you appreciate the volume of experiments in the manuscript.
>
> For calibration purposes, we’d like to note that the ICLR 2025 rubric differs slightly from previous similar conferences. For example:
>
> - To indicate "Accept", the NeurIPS 2024 rubric says to use 7 whereas the ICLR 2025 rubric says to use 8
> - To indicate "Strong Accept", the NeurIPS 2024 rubric says to use 9 whereas the ICLR 2025 rubric says to use 10
>
> To address the concerns you raised:
>
> > the paper starts with MULTIVARIATE GAUSSIAN MODELING, can the findings still hold for other model, like VAE, diffusion model, or Transformer. Somehow, I feel the paper make too strong claim since there are many generative model.
>
> 1.  We began with multivariate Gaussian modeling because it featured prominently in Shumailov, a Nature publication with ~250 citations. We felt studying such a prominent setting was an appropriate place to begin. Additionally, we wanted both theoretical and experimental results to complement one another, and multivariate Gaussian modeling is one setting where both are possible. To the best of our knowledge, VAEs, diffusion models and transformers are currently analytically intractable.
> 2.  To emphasize, we don’t study JUST multivariate Gaussians. We study many additional generative models, including instruction finetuning of transformer-based language models, kernel density estimation, linear regression and pretraining of transformer-based language models. This set of model settings includes both unsupervised and supervised learning, as well as both autoregressive and i.i.d. data.
> 3.  If you feel like specific generative models are missing that would strengthen our paper’s results, please let us know. We would be happy to run additional experiments (assuming such experiments are within our compute budget and can be run within the ICLR discussion period).
>
> > the reprodubility is unclear at this time..
>
> We ran 5 sets of experiments, each with at least three seeds, and provided error bars for our results.  All of our results were consistent with our claims. We would be happy to provide you with an anonymized version of our code.
>
> Could you please elaborate on what you feel would strengthen the reproducibility of our paper? Assuming the changes are feasible and within reason, we will make such changes.
>
> > the math proof in Appendix is mainly for Guassina model, it is unclear how it can generalize to more generative models.
>
> Our appendix has two mathematical proofs, one for Gaussian modeling and another for Kernel Density Estimation. Given the difficulty the field has with analyzing even 3-layer networks, rigorous proofs for more complicated generative models (e.g., diffusion models) of model collapse are likely beyond the field’s current mathematical toolkit. If you have suggestions for how to generalize the proofs, we would be very keen to hear your suggestions.

---

> ### Author Response · Authors · 2024-11-24
> **Rebuttal to Reviewer xzu3 (part 2)**
>
> We have updated the manuscript with several improvements:
> - We added a related works section.
> - We included additional experiments in the SFT setting with Gemma-2-9B and Gemma-2-27B (Figure 3), which we believe will further allay your concerns about reproducibility.
>
> In light of these improvements and our earlier rebuttal, we hope that you will consider increasing your score.

---

> > ### Author Response · Authors · 2024-11-27
> > **Request for Feedback from Reviewer xzu3**
> >
> > Thank you for your helpful suggestions on our paper.
> >
> > Given that today is our final opportunity to update the manuscript, we would appreciate any further guidance on whether your concerns have been addressed by our rebuttal, added experiments, and revised manuscript.  If they have, please consider raising your score.  If they have not, please let us know what else we can do to assuage your remaining concerns.

---

> > > ### Author Response · Authors · 2024-12-03
> > > **Final Reminder to xzu3**
> > >
> > > Dear xzu3,
> > >
> > > We made an effort to address your concerns about reproducibility and the fact that we study statistical models as special cases.  We have not received a response or an updated score from you.  Since we are no longer able to comment starting tomorrow, we wanted to take this opportunity to ask you to reconsider our paper’s score in light of the rebuttal that we wrote and the considerable improvements that we have made to the manuscript.
> > >
> > > Thank you,
> > >
> > > The Authors

---

### Official Review · Reviewer_MNCh · 2024-11-05

**Soundness:** 3
**Presentation:** 3
**Contribution:** 3
**Rating:** 8
**Confidence:** 2

**Summary:**

Update: I appreciate the authors efforts to address my concerns, and thus would like to raise my score. Ideally, I would change it to a 7 b/c I feel it does deserve acceptance but not spotlight/oral.

This paper studies how training on synthetically generated data in the model-data feedback loop. In particular the authors claim to focus on a more nuanced setting than previous papers on the topic by considering settings where (i) parts of the real-data and synthetic data are used together (instead of discarding the real-data entirely in favor of synthetic), (ii) only a fixed budget is available for updating the model, and (iii) consider how important the proportion vs. cardinality is when updating the model. The authors demonstrate that keeping some of the real data can help prevent collapse, that the right amount of synthetic + real data can improve performance, and in some cases, no amount of synthetic data is outperform real data. The authors provide some proofs to demonstrate their claims too.

**Strengths:**

- *Important Problem Setting.* The problem of model collapse is a pressing problem and I found the authors investigation into the more nuanced issues of the model-data feedback loop compelling: that likely parts of synthetic and real data will be aggreagated, the budget assumptions etc.

- *Thorough Experiments.* The authors conduct well-designed experiments on a variety of models and settings, including Gaussian models, kernel density estimators, linear regression and language models on a variety of datasets. While some of these models and datasets are fairly simple, they nonetheless help suport the author's claims.

- *Theoretical Insights.*  In addition to the empirical insights, the authors are able to proud some theoretical guarantees. This helps complement the empirical results with simpler models.

**Weaknesses:**

- * Novelty.* The novelty of this work is fairly limited since it builds quite a bit on existing work. It starts by addressing some of the claims from Gertgrasser et al, and then uses the model/settings in Shumailov et al. The settings the authors tried out (not replacing data en-masse. etc) are important but do seem a bit incremental conceptually.

- *More Real-settings.* The authors motivate their work a fair bit from the perspective of language models. I would like to have seen more experiments on this setting under a couple of different model strengths and synthetic data quality.

**Questions:**

- Can the authors please clarify their novelty a bit more?
- Can the authors please discuss how they expect their results to hold when training significantly larger models, or changing the quality of the synthetic data?

---

> ### Author Response · Authors · 2024-11-14
> **Rebuttal to Reviewer MNCh**
>
> Thank you for reviewing our paper. We are grateful you found our experiments thorough and our theoretical insights a positive contribution.
>
> For calibration purposes, we’d like to note that the ICLR 2025 rubric differs slightly from previous similar conferences. For example:
>
> - To indicate "Accept", the NeurIPS 2024 rubric says to use 7 whereas the ICLR 2025 rubric says to use 8
> - To indicate "Strong Accept", the NeurIPS 2024 rubric says to use 9 whereas the ICLR 2025 rubric says to use 10
>
> To address the concerns you raised:
>
> > Novelty.* The novelty of this work is fairly limited since it builds quite a bit on existing work. It starts by addressing some of the claims from Gertgrasser et al, and then uses the model/settings in Shumailov et al. The settings the authors tried out (not replacing data en-masse. etc) are important but do seem a bit incremental conceptually.
>
>
> Hopefully we can clarify why we feel this manuscript contains multiple novel insights.
>
> 1.  The first third of our manuscript does indeed test whether the data paradigm claims of Gerstgrasser hold in the model settings of Shumailov. However, we feel that a scientific contribution like this (i.e., testing whether a prediction holds) is the fundamental basis of science and should not be dismissed as lacking novelty. Predictions need to be tested!
> 2.  The experiments in the first third of our manuscript significantly extend beyond either original paper. Gerstgrasser considers the settings of language model pretraining, linear regression, VAEs, and diffusion. In contrast, we consider Gaussian mean/covariance estimation, KDEs, and SFT of language models.  Moreover, Gerstgrasser experiments on TinyStories only ran 5 model-fitting iterations, whereas we go to over 40 model-fitting iterations for Replace and to 10+ iterations for Accumulate.
> 3.  The second third of our manuscript introduces a middle ground for how data are treated between Accumulate and Replace. This data paradigm might be conceptually simple, but it has significant real world implications and thus needs to be studied carefully. While accumulating data avoids model collapse, training on all accumulating data rapidly increases the cost of model training. Thus, it is highly unlikely that future large-scale deep generative models will be trained on 10 quadrillion or 100 quadrillion tokens, and it is instead far more likely that web-scale datasets will be collected and filtered to a smaller size. Section 3 and Figure 4 provide a careful examination of 5 different generative modeling settings to study this crucial middle ground.
> 4.  The final third of our manuscript moves far beyond prior work to tease apart the relative importance of proportionality versus cardinality of real and synthetic data on SFT performance, which previous papers on model collapse did not directly compare, but turns out to be crucial. We also provide insights into the value of synthetic data, discovering that when the number of real data is insufficient, a specific amount of synthetic data can improve model performance.
>
>
> > More Real-settings. The authors motivate their work a fair bit from the perspective of language models. I would like to have seen more experiments on this setting under a couple of different model strengths and synthetic data quality.
>
> What experiment(s) would you most like to see? We are happy to run additional experiments, but our compute budget is finite and the review timeline is short, so we want to make sure we prioritize the experiments you think would be most valuable. For instance, we can extend the SFT results (Figure 3) to include other language models, e.g., Gemma 2 9B and Gemma 2 27B.

---

> ### Author Response · Authors · 2024-11-24
> **Rebuttal to Reviewer MNCh (part 2)**
>
> We have updated our manuscript with the following improvements:
> - We included a related works section.
> - In accordance with your suggestion that we should explore different model strengths that produce different quality synthetic data, we included preliminary experiments with Gemma-2-9B and Gemma-2-27B in the SFT setting.  Note that model collapse still occurs in the replace setting, though it appears to happen more slowly with larger models.  These can be found in Figure 3.
>
> If this was not the experiment that you were looking for, please let us know so that we can correct course.
>
> Otherwise, we hope that you'll consider raising your score in light of these improvements.

---

### Public Comment · ~Julia_Kempe1 · 2024-11-26
**Public comment by 9 members of the model collapse community**

1/2

Dear Reviewers and AC,

In this public comment, we would like to respectfully point out that this paper fails to adequately acknowledge all perspectives (and large amounts of related works) on model collapse and, by extension, on mitigating it. As such, their definition and approach to model collapse offers at best a partial view. We feel it seriously misrepresents other works, which should not happen when the purported goal is to unify fractured literature (see point 1 below).

Claims that warrant a nuanced viewpoint are stated unequivocally: for instance (line 21): *".. previous work claimed that model collapse is caused by replacing all past data with fresh synthetic data at each model-fitting iteration and that collapse is avoided by instead accumulating data across model-fitting iterations"* hardly summarizes the status of the literature, which contanis a wealth of explanations and mechanisms.

Moreover, we want to point out omission of prior work that already studied parts of "new contributions" of this paper, namely the benefits of synthetic data when real data is scarce (see point 3 below).
We also highly comment reviewer *BKEn* for noting that the contribution of the presented work is essentially additional validation of the findings of previous work (Gerstgrasser et al., 2024). Where novel theory would have been welcome (subsampling from a mixture of real and synthesized data to compare apples to apples in terms of dataset size), it seems lacking (point 2 below). In detail.

1. **Clarification and Definition of Model Collapse.** We appreciate the effort in addressing and attempting to unify the fractured literature on the perils and promises of synthetic data. However, to achieve this goal effectively, the paper must clearly define "model collapse" and elucidate what avoiding model collapse means. The definition of model collapse varies across prior works:
- Shumailov et al. (2024) define model collapse as the phenomenon where “a degenerative process affecting generations of learned generative models, in which the data they generate end up polluting the training set of the next generation.” They continue to define that model collapse is not just that the final model is catastrophically broken, it is also gradual loss of representativeness and improbable events.They called them *stages* in the paper.
- Model collapse is not only about finiteness of synthetic or real data, it is really the interaction of finiteness and overregularization akin to Mobahi et al. (https://arxiv.org/abs/2002.05715), which can happen in kernel density estimation, as shown in Shumailov et al. (Nature, 2024).
- Dohmatob et al. (2024b) identify model collapse as the alteration of scaling law curves, where training on synthetic data breaks the scaling law compared with the scaling law of training on real data. Continuous retraining leads to a n-fold divergence. Avoiding model collapse in this context involves achieving a scaling law curve comparable to that of real data.

Gerstgrasser et al. (2024) and this paper conceptualize model collapse as a scenario where continuous retraining on synthetic data results in diverging performance, with avoiding collapse equated to maintaining a non-diverging performance. These definitions are all different and none seems more valid than any other. As a consequence there is a body of works taking different viewpoint on what "avoiding model collapse" means. Let us just list:

- Zhang, Jinghui, et al. "Regurgitative training: The value of real data in training large language models." arXiv preprint arXiv:2407.12835 (2024). "All three strategies can improve the performance of regurgitative training to some extent but are not always able to fully close the gap from training with real data."
- Alemohammad, Sina, et al. "Self-improving diffusion models with synthetic data." arXiv preprint arXiv:2408.16333 (2024) also terms preventing MaDness (collapse) as whether we could improve beyond the generator of the syntehtic data.
- Peterson, Andrew J. "AI and the Problem of Knowledge Collapse." arXiv preprint arXiv:2404.03502 (2024) also calls the prevention of collapse to be preserve an accurate vision of the truth for the community.
- Feng, Yunzhen, et al. "Beyond Model Collapse: Scaling Up with Synthesized Data Requires Reinforcement." arXiv preprint arXiv:2406.07515 (2024) also defines avoiding model collapse as attaining performance beyond the generator that synthesize data.
- Wang, Ze, et al. "Bias Amplification: Language Models as Increasingly Biased Media." arXiv preprint arXiv:2410.15234 (2024) treats mixing and accumulation as mitigation methods.

(cont. in 2/2)

---

> ### Public Comment · ~Julia_Kempe1 · 2024-11-26
> **Public comment by 9 members of the model collapse community (cont. 2/2)**
>
> 2/2
>
> - Seddik et al. (https://arxiv.org/pdf/2404.05090): in a very relevant work, they presented a rather nice analysis of both fully synthetic and partially synthetic loops, and clearly state (with exact error bounds): "highlights that the amount of synthetic data should be exponentially smaller compared to real data in order to ensure that p(m) remains close to p(1)."
>
> To truly unify the literature, as the paper claims it does, it should present a detailed discussion of these varying perspectives and their implications. Without this, the goal of unifying the "fractured" literature remains shockingly unmet.
>
> 2. **Theoretical Analysis of Accumulate-Subsample Strategy.** A series of accumulate--subsample experiments for the Gaussian setting has already been performed in the Nature paper of Shumailov et al. (2024) (Fig 4 and 5 of the supplementary) and even there the model collapse drift keeps on happening, although slower.
> Thus we wonder, to add new contributions, could the authors theoretically demonstrate that the accumulate-subsample approach prevents divergence? While the results in the “Pretraining of Language Models on TinyStories” section show that replacing data leads to divergence, the accumulate-subsample strategy also shows an increasing trend. Providing a theoretical foundation for why this approach should avoid divergence would strengthen the argument and clarify its effectiveness (and provide theory beyond what is already contained in (Gerstrgrasser et al. 24)).
>
> 3. **Interaction Between Real and Synthetic Data.** The interaction between real and synthetic data is theoretically analyzed in Dohmatob et al. (2024b), specifically in Corollary 3.3 and Figure 5. Their results suggest that synthetic data is beneficial when real data is scarce but becomes detrimental when large amounts of real data are available. This relationship is similar to the articulated principle in this paper that “the value of synthetic data for reducing model test loss depends on the absolute quantity of real data.” The prior result should be acknowledged.
>
> **We hope our comment can allow for a nuanced evaluation of this paper. Thank you.**
>
> Signed by:
>
> Sina Alemohammad - Rice University
>
> Quentin Bertrand - INRIA
>
> Joey Bose - University of Oxford
>
> Elvis Dohmatob - Mila, Concordia & MetaAI
>
> Yunzhen Feng - New York University
>
> Gauthier Gidel - University of Montreal & Mila
>
> Julia Kempe - New York University & MetaAI
>
> Ilia Shumailov - Google
>
> Zakhar Shumailov - University of Cambridge

---

> ### Author Response · Authors · 2024-11-27
> **Author Response to "Public comment by 9 members of the model collapse community" (Part 1)**
>
> Thank you for your comments and questions regarding our ICLR submission.
>
> > Clarification and Definition of Model Collapse. We appreciate the effort in addressing and attempting to unify the fractured literature on the perils and promises of synthetic data. However, to achieve this goal effectively, the paper must clearly define "model collapse" and elucidate what avoiding model collapse means. The definition of model collapse varies across prior works:
>
> > These definitions are all different and none seems more valid than any other.
>
> We agree that the literature employs varying notions of “model collapse” which can lead to confusion. We take model collapse by its literal meaning: that model (performance) crumbles/disintegrates/fails catastrophically when models are trained on synthetic data. A concurrent ICLR submission similarly defines model collapse as “a critical degradation in the performance of AI models.” We prefer this notion because it justifies why society and researchers care so intently about model collapse, i.e., the threat that future generative models are imminently and fatally
> doomed. After all, the phenomenon is called “model collapse”, not “model slow degradation” or “model multiple scaling laws”! We will explicitly include this notion of model collapse in the manuscript, as well as why we prefer this particular notion.
>
> > To truly unify the literature, as the paper claims it does, it should present a detailed discussion of these varying perspectives and their implications. Without this, the goal of unifying the "fractured" literature remains shockingly unmet.
>
> We appreciate this feedback and the opportunity to clarify. Our goal is not to present a detailed discussion of varying perspectives. Rather, we are unifying the fractured experimental methodologies in the literature whereby each paper considers its own choices of generative models, its own choices of how data are handled, its own notions of whether model collapse has or has not occurred.  We will update the abstract to clarify.
>
> The first two thirds of our manuscript presents a systematic, side-by-side evaluation of multiple generative models under different data accumulation/replacement paradigms to demonstrate the different behaviors of model loss on real test data. By providing this consistent comparative lens, we made the patchwork of methods and results in the literature more navigable and cohesive, with more clear conclusions.
>
> > We also highly comment reviewer BKEn for noting that the contribution of the presented work is essentially additional validation of the findings of previous work
>
> The value we added in the first third of our manuscript is performing an apples-to-apples comparison that did not previously exist, as we explained to Reviewer BKEn. The previous papers used different generative models and different data paradigms, meaning the field lacked a consistent side-by-side comparison.
>
> > Claims that warrant a nuanced viewpoint are stated unequivocally: for instance (line 21): ".. previous work claimed that model collapse is caused by replacing all past data with fresh synthetic data at each model-fitting iteration and that collapse is avoided by instead accumulating data across model-fitting iterations" hardly summarizes the status of the literature, which contains a wealth of explanations and mechanisms.
>
> To clarify,
>
> 1. The quoted Line 21 is taken from the abstract. Delving into “a wealth of explanations and mechanisms” in a one paragraph abstract is not feasible.  We expand on the state of model collapse research in the body of the paper.
>
> 2. The “previous work” refers to Gerstgrasser et al. 2024. We did not explicitly state the reference in the abstract because (i) abstracts do not typically contain references and (ii) an unfamiliar reader might be unfamiliar with that prior work.
>
> 3. Regardless, our goal in the abstract was to succinctly convey the state of the field, and we believe that this sentence is a reasonably accurate description of the literature (or as accurately as one can manage in a sentence).  If we find a better succinct description, we will update in our next version of the manuscript.

---

> ### Author Response · Authors · 2024-11-27
> **Author Response to "Public comment by 9 members of the model collapse community" (Part 2)**
>
> > A series of accumulate--subsample experiments for the Gaussian setting has already been performed in the Nature paper of Shumailov et al. (2024) (Fig 4 and 5 of the supplementary) and even there the model collapse drift keeps on happening, although slower.
>
> We were aware of these experiments. We found it difficult to draw conclusions from Figs 4 & 5 of the supplementary for a few reasons:
>
> - To the best of our ability, we could not find the methodology for Figs 4 & 5, making interpreting the results difficult.
> - As best as we can tell, Figs 4 & 5 are not discussed in the main text nor in the supplement.
> - The results in Figs 4 & 5 are, in our opinion, unclear. We do not agree with your claim that “even there, the model collapse drift keeps on happening”. Rather, we feel that Fig 5 is noisy and seems to suggest that the processes stabilizes.
> - As best as we can tell, only a single seed was run, meaning we do not know whether Figs 4 & 5 showcase “typical” behavior.
>
> In contrast, our Figure 4 presents a digestible side-by-side view of Gaussian model behavior under Replace, Accumulate-Subsample, and Accumulate paradigms, with 100 seeds per hyperparameter configuration. We also visualize the Gaussian model behavior alongside 4 additional generative models, enabling easy comparison. We believe this synoptic visualization provides the field with a clearer picture of the key results, which are not visualized, communicated or emphasized in Shumailov et al. 2024.
>
> > The interaction between real and synthetic data is theoretically analyzed in Dohmatob et al. (2024b), specifically in Corollary 3.3 and Figure 5. Their results suggest that synthetic data is beneficial when real data is scarce but becomes detrimental when large amounts of real data are available.
>
> We believe that this is not a faithful characterization of what Dohmatob et. al. 2024b show. Rather, their Figure 5 shows that even with large real datasets (dark line), additional synthetic data (moving rightward on x-axis) continues to reduce test error, contradicting your statement that “synthetic data [...] becomes detrimental when large amounts of real data are available”. The authors even write: "Figure 5 illustrates how AI data can boost performance, up to a certain point, when its benefits plateau."
>
> > As a consequence there is a body of works taking different viewpoint on what "avoiding model collapse" means.
>
> Per ICLR guidelines (https://iclr.cc/Conferences/2025/FAQ), these works count as contemporaneous, and not citing them is not grounds for rejection: “We consider papers contemporaneous if they are published within the last four months. [...]  Authors are encouraged to cite and discuss all relevant papers, but they may be excused for not knowing about papers not published in peer-reviewed conference proceedings or journals, which includes papers exclusively available on arXiv.”
>
> To the best of our ability to discern, of the 5 suggested papers (Zhang et al. 2024, Alemohammad et al. 2024, Peterson 2024, Feng et al. 2024, Wang et al. 2024):
>
> - 4 of the 5 are preprints and 1 is a workshop paper.
> - 4 of the preprints were posted within the past 4 months.
> - **1 of the preprints was posted after the ICLR submission deadline.**
>
> However, we agree that engaging with these works, even if not strictly required, could enhance our paper. In our revision, we will include a discussion of these papers before the revision period ends, clarifying how our work aligns or differs from their findings. We believe this addition will contribute to a more comprehensive picture of the field.
>
> In summary, we appreciate your comments and the issues you raised. We look forward to the opportunity to develop and share a new draft of our manuscript with the research community.

---

> > ### Public Comment · ~Julia_Kempe1 · 2024-12-03
> > **A quick final response**
> >
> > Without wanting to extend this discussion unduly beyond our first comment by repeating our points, and to abuse the patience of the reviewers, we want to thank the authors for their engagement. Given our disagreements remain, we only wish to clarify one technical point:
> >
> > Corollary 3.3 and Figure 5 in Dohmatob et al. (2024b) provide a precise characterization of the conditions under which synthetic data proves beneficial. Specifically, Corollary 3.3 demonstrates that when $T_{\text{real}} \ll k^{\beta}$, an abundance of synthetic data can enhance performance. This result illustrates: "How AI data can boost performance up to a certain point, after which its benefits plateau." However, when the quantity of real data exceeds this threshold, synthetic data ceases to contribute to performance gains from Theorem 3.2. All combinations shown in Figure 5 fall under the condition that $T_{\text{real}} < k^{\beta}$. Once $T_{\text{real}}$ becomes sufficiently large, adding synthetic data no longer leads to performance improvements.
> >
> > Thank you.

---

### Author Response · Authors · 2024-12-02
**General Response to the AC and Reviewers**

Dear AC and Reviewers,

Thank you for reading our paper and offering helpful suggestions.  Because the discussion period is ending, and we have not received responses or updated scores from two reviewers, we would like to summarize the main concerns of the reviewers and how we addressed them.

__Larger models__: All reviewers questioned how our results would change under different model sizes and/or strengths, since larger models would likely improve synthetic data quality.  To address this concern, we ran additional SFT experiments with Gemma-2-9B and Gemma-2-27B (Figure 3).  These experiments suggest that our results hold regardless of model scale, but larger models collapse more slowly under the replace paradigm.

__Reproducibility__: Reviewer xzu3 states “the reprodubility is unclear at this time.”  We have provided 5 sets of experiments run across 3 seeds with error bars.  The results are all consistent with each other.  Moreover, for the final version, we have agreed to release the Github for the project, which provides access to all of our models, sweep settings, and runs.  In our opinion, objections to the paper on the grounds of reproducibility are unfounded.

__Novelty__: Two reviewers (MNCh, BKEn) raised concerns about the novelty of our work.  Originally, Gerstgrasser et. al. postulated that accumulating data avoids collapse.  However, their theoretical and experimental settings were not extensive enough to support the universality of the claim.  We provided theoretical evidence in 2 new settings and experimental evidence in 3 new settings to support the generality of Gerstgrasser’s claim.  Note that previous model collapse works studied some of the same modelling settings, but they lower-bounded the proportion of real data (Bertrand et. al. (2024), Dohmatob et. al. (2024b)), whereas we allow it to go to 0.  We also replicated Gerstgrasser’s pretraining experiment with more training iterations and provided evidence that in low data settings, an optimal amount of synthetic data can improve model  SFT performance.  Finally, we explored a new compute-limited paradigm that we call “accumulate-subsample.”  Accumulate-subsample represents a middle-ground between replace and accumulate.  The extensive new theoretical and experimental results that we have provided justify our work’s novelty.

__Proof Assumptions__: Two reviewers (9sJL, xzu3) object to our proof settings, noting that theorems about Gaussian estimation and KDEs do not generalize to LLMs.  Currently, the field does not know how to analytically probe general LLMs, and using Gaussians and KDEs to gain intuition about language models has a strong precedent in Shumailov et. al. (2023).  Shumailov explored these settings under the replace paradigm to provide evidence of collapse, so we chose to use the same settings under the accumulate paradigm to show that collapse does not occur.  As a result, our lack of analytic proofs for deep language models should not be viewed as a shortcoming of our paper.

__Related Works__: Reviewer BKEn commented that we were missing a related works section.  We have added one, and will revise it in accordance with his suggestions in our final version.

__Conclusion__:
Our paper has significantly improved through our efforts to address the reviewers’ concerns.  We hope that the remaining reviewers will respond and acknowledge these improvements.  If they do not respond, __we implore the AC to consider that our revisions have obviated most of the objections posed to our paper.__

Sincerely,

The Authors

---

### Meta-Review · Area_Chair_3rLz · 2024-12-29

**Metareview:**

This paper studies model collapse, an important shortcoming of generative models when they are trained on synthetic data. The major concern that reviewers have, including public reviewers from the model collapse community, is that the paper is of limited novelty and doesn't fully contextualize itself in the broader model collapse literature. The reviewers judged this paper fails to properly discuss key papers making similar contributions. There has been a long exchange between the authors and the reviewers. My recommendation for rejection is based on reading the paper and on the authors' failure to address all the reviewers' concerns when it comes to previous work.

**Additional Comments On Reviewer Discussion:**

The authors have provided a rebuttal and have engaged with the reviewers. The authors did not address the final comment of the public reviewers from the model collapse community.

---

### Decision · Program_Chairs · 2025-01-22

Reject